# Leveraging functional genomic annotations and genome coverage to improve polygenic prediction of complex traits within and between ancestries

Zhili Zheng [1,2,3] ✉, Shouye Liu[1], Julia Sidorenko [1], Ying Wang [1], Tian Lin [1], Loic Yengo [1], Patrick Turley [4,5], Alireza Ani [6,7], Rujia Wang [6], Ilja M. Nolte [6], Harold Snieder [6], LifeLines Cohort Study*, Jian Yang [8,9], Naomi R. Wray [1,10], Michael E. Goddard[11,12], Peter M. Visscher [1,13] & Jian Zeng [1] ✉

We develop a method, SBayesRC, that integrates genome-wide association study (GWAS) summary statistics with functional genomic annotations to improve polygenic prediction of complex traits. Our method is scalable to whole-genome variant analysis and refines signals from functional annotations by allowing them to affect both causal variant probability and causal effect distribution. We analyze 50 complex traits and diseases using ~7 million common single-nucleotide polymorphisms (SNPs) and 96 annotations. SBayesRC improves prediction accuracy by 14% in European ancestry and up to 34% in cross-ancestry prediction compared to the baseline method SBayesR, which does not use annotations, and outperforms other methods, including LDpred2, LDpred-funct, MegaPRS, PolyPred-S and PRS-CSx. Investigation of factors affecting prediction accuracy identifies a significant interaction between SNP density and annotation information, suggesting whole-genome sequence variants with annotations may further improve prediction. Functional partitioning analysis highlights a major contribution of evolutionary constrained regions to prediction accuracy and the largest per-SNP contribution from nonsynonymous SNPs.

Polygenic scores (PGSs) for complex traits are playing increasingly important roles in research and medical applications of the fast-growing genomic data from genome-wide association studies (GWASs)[1]. PGSs are used to provide evidence of polygenic adaptation of populations to different environments[2], explore putative causal relationships between traits[3], improve cost and efficiency of clinical trials[4] and, perhaps most importantly, identify individuals with high genetic risk of complex diseases[5–10], which opens up opportunities for preventative medicine, early intervention and personalized treatment[11–13]. However, the clinical application of PGSs is currently limited by the modest prediction accuracy for most complex diseases. Moreover, a substantial loss of prediction accuracy is observed when applying PGSs across ancestries[14–20].

The prediction accuracy of PGSs depends on the selection of SNPs in the model and the estimation of their effects. For cross-ancestry prediction, the accuracy further depends on the extent to which the linkage disequilibrium (LD) in the GWAS population matches that in the target population. Although mounting evidence suggests that common causal variants are shared across ancestry groups[20,21], selecting

A full list of affiliations appears at the end of the paper. *A list of authors and their affiliations appears at the end of the paper.
✉e-mail: zhili.zheng@broadinstitute.org; j.zeng@uq.edu.au

**a**

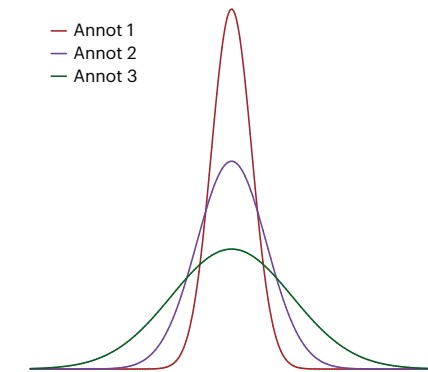

**Fig. 1 | Characteristics of functional annotation data. a**, Functional annotations provide orthogonal information that helps to distinguish the causal variant (CV) from the SNP in perfect LD with it. However, when the causal variant is not observed, its effect can be captured through LD by an SNP that has a different annotation from the causal variant, resulting in a mismatch between effect size and annotation category (denoted by 'Annot'). **b**, Functional categories can differ in both the proportion of causal variants and the distribution of causal effect sizes, either of which can lead to an enrichment or depletion in per-SNP heritability in a functional category.

these variants only in the PGS model is challenging because, due to the action of negative selection[22–24], complex traits are affected by many common causal variants, with vanishingly small effect sizes and in LD with non-causal SNPs in their vicinity.

Functional genomic annotations can be used to distinguish likely causal SNPs from non-causal SNPs in high LD with them[25], thereby improving polygenic prediction[15,26–29]. The idea of using functional annotations to improve prediction was first proposed in livestock genetics through a method called BayesRC[30] based on individual-level data. Recent methodological development in human genetics have allowed the integration of GWAS summary-level data with annotations for polygenic prediction, including AnnoPred[27], LDpred-funct[28], MegaPRS[29] and PolyPred[15]. However, there are limitations in these methods. First, it is common to consider only a subset of common variants (for example, SNPs from a genotyping array or the HapMap3 panel[31]) due to computational feasibility. This practice may potentially be problematic, as SNP markers can capture the effects of unobserved causal variants through LD but may not share the same annotation with the causal variants (Fig. 1a). Second, these methods are all stepwise and depend on the estimated per-SNP heritability enrichment for each annotation[32] as input data in the initial step. This enrichment can result from variations in either the proportion of causal variants or the distribution of effect sizes across annotation levels or categories[30,33] (Fig. 1b). Notably, none of these methods explicitly account for the two sources of information in a unified model that simultaneously fit GWAS data and functional annotations.

Here, we propose a new method, SBayesRC, that addresses these limitations by analyzing all imputed common SNPs simultaneously using an efficient algorithm, refining the annotation information using a hierarchical multicomponent mixture prior and estimating all parameters jointly from the data using a full Bayesian learning machinery. We apply our method to 50 complex traits, with up to 10 million imputed

common SNPs and 96 functional annotations. We consider both within European ancestry and cross-ancestry prediction using datasets from multiple biobanks and large consortia, comparing with the best methods in the literature (Extended Data Table 1). Moreover, we investigate factors that affect prediction accuracy and consider connections between the genetic architecture of functional categories and their contributions to prediction accuracy.

## Results

### Method overview

SBayesRC extends SBayesR[34] to incorporate functional annotations and allows for the joint analysis of all common SNPs in the genome. It only requires summary statistics from GWAS and LD correlations from a reference sample as input data, outputting joint SNP effect estimates for the PGS calculation. In addition, it provides posterior inclusion probabilities (PIP) for SNPs as measures of trait associations and estimates of functional genetic architecture parameters like SNP-based heritability and polygenicity associated with the functional annotations.

Compared to other methods, SBayesRC has two unique features. First, it utilizes a low-rank model to efficiently fit all common variants and better model the LD between them (Methods, Extended Data Fig. 1a and Section 1 of the Supplementary Note). Based on the eigen-decomposition on quasi-independent LD blocks in the human genome[35], the low-rank model refines the signals in GWAS summary statistics by collapsing information from SNPs in high LD, leading to significantly improved computational efficiency and enhanced robustness to LD differences between GWAS and reference samples (Section 2 of the Supplementary Note). Second, a multicomponent annotation-dependent mixture prior is used to better model the distribution of SNP effects and to learn both annotation parameters and SNP effects from the data (Methods, Extended Data Fig. 1b and Section 5 of the Supplementary Note). By allowing the annotations to affect the

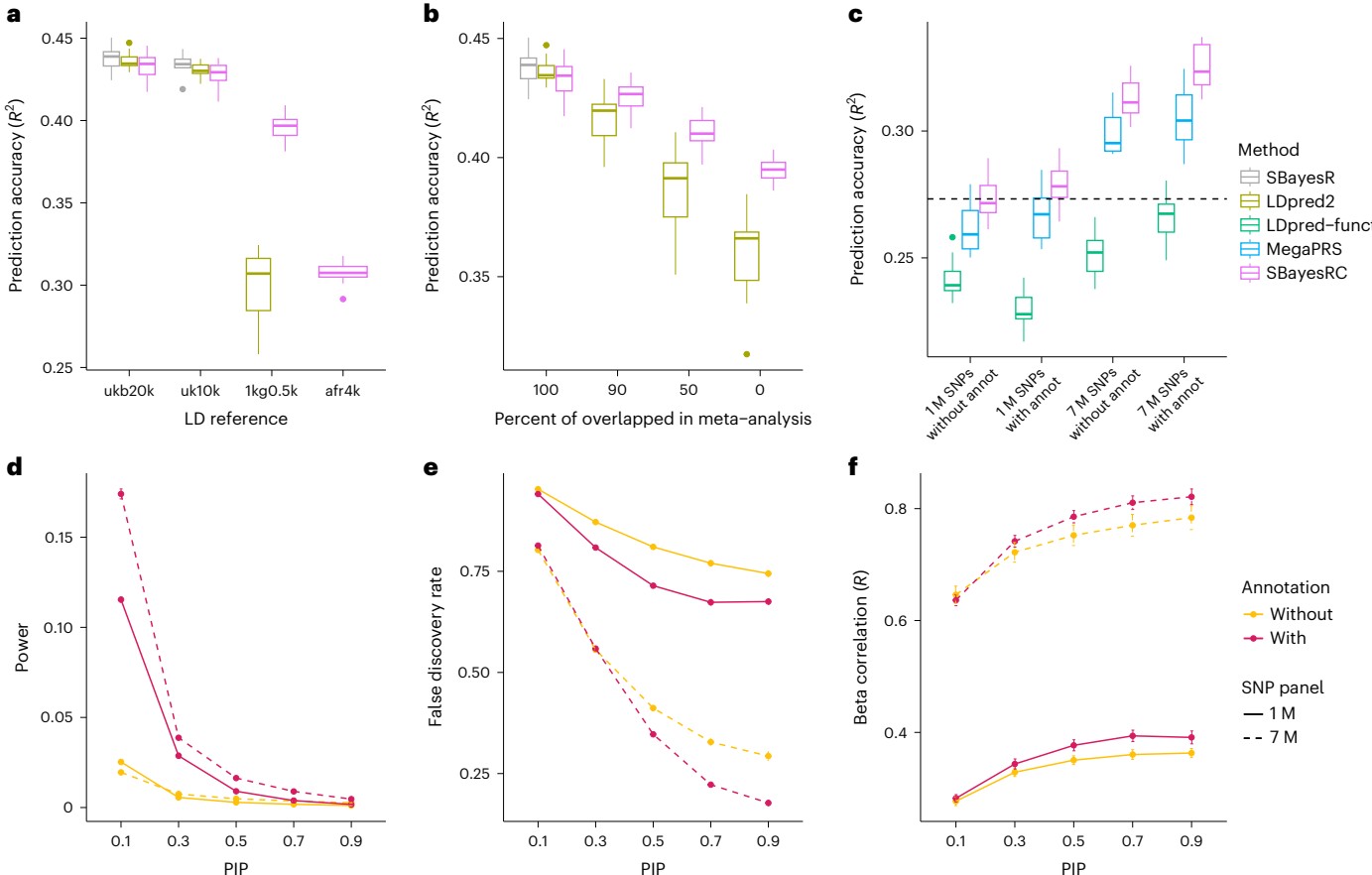

**Fig. 2 | Assessing the performance of different methods by simulations.**
**a**, Robustness of SBayesRC to the choice of LD reference (ukb20k, a random sample of 20,000 unrelated individuals of European (EUR) ancestry from the UKB; uk10k, 3,642 unrelated EUR individuals from the UK10K dataset; 1kg0.5k, 494 unrelated EUR individuals from the 1000 Genomes Project; afr4k, a random sample of 4,000 unrelated individuals of African ancestry from the UKB).
**b**, Robustness of SBayesRC to the unequal per-SNP sample sizes in the meta-analysis. **c**, The prediction $R^2$ from SBayesRC, LDpred-funct and MegaPRS with different SNP densities and with or without annotations. The dashed line shows the prediction $R^2$ from the benchmarking method SBayesR using HapMap3 SNPs without annotations. **d**, Power of identifying causal variants using SBayesRC with

or without high-density SNPs or annotation data. **e**, False discovery rate (FDR) of identifying causal variants using SBayesRC with or without high-density SNPs or annotation data. **f**, Correlations between the SBayesRC estimated and true effect sizes at SNPs with posterior inclusion probability (PIP) greater than a threshold. Results were from simulations ($n = 10$ independent replicates) with trait heritability $h^2 = 0.5$ (the upper bound of the prediction accuracy). See Extended Data Figs. 2 and 3 and Supplementary Figs. 3–5 for results from the simulation with $h^2 = 0.1$. Each box plot in **a**–**c** shows the spread of data; the line is the middle (median), the box covers the middle half (IQR), the whiskers extend to 1.5 times the IQR, and dots show outliers. Data in **d**–**f** are presented as mean values (center point) ± standard error of the mean (s.e.m.) (error bar) in each PIP bin.

probability that SNPs are causal variants and the probability distribution of their effect sizes, SBayesRC can better capture the causal effects if the distributions of effect sizes truly differ between annotations. The method has been implemented in an R package and the software *GCTB*[23] (see Code availability).

**Genome-wide simulation based on real genotypes and annotation data**
We first calibrated the low-rank model with simulation in HapMap3 SNPs to determine the best parameter setting for polygenic prediction (Section 9 of the Supplementary Note). We then tested our method under two common issues encountered in practice: (1) differences in LD between GWAS and LD reference datasets and (2) unequal GWAS sample sizes across SNPs (Section 11 of the Supplementary Note), in comparison to two state-of-the-art methods using summary statistics, LDpred2 (ref. 36) and SBayesR[34]. For all methods, a decrease in prediction accuracy was observed when the LD reference sample size was too small relative to the GWAS sample size, indicating an important variation in LD by chance (Fig. 2a and Extended Data Fig. 2). However, SBayesRC (without annotation) preserved more prediction accuracy than the other methods. In an extreme case where LD correlations were estimated using individuals of

African ancestry, SBayesRC achieved a preservation of ~70% prediction accuracy, whereas SBayesR and LDpred2 (default settings) were unable to reach convergence. Regarding the scenario of unequal per-SNP sample sizes, as the proportion of overlapped SNPs decreased, SBayesR more frequently failed to converge, and LDpred2 exhibited a faster rate of decrease in prediction accuracy compared to SBayesRC (Fig. 2b). It is noteworthy that the impact of model misspecification was mostly absorbed in the nuisance residual variance in SBayesRC, resulting in less bias in the genetic architecture parameters, such as SNP-based heritability and polygenicity, compared to LDpred2 (Extended Data Fig. 3).

We next assessed the benefits of using functional annotation data by expanding the simulation to include 7,356,518 imputed common SNPs and incorporating functional annotations to simulate the causal effects (Methods). As expected, the result demonstrated a significant improvement in prediction accuracy when using more SNPs and/or annotation data in SBayesRC (Fig. 2c). Compared to using 1 M HapMap3 SNPs, using all 7 M SNPs led to a 14.4% increase in prediction accuracy (calculated as $(R^2_{7M} - R^2_{1M})/R^2_{1M}$, where $R^2$ is the prediction $R^2$ in the validation sample). Compared to the no-annotation model, the model incorporating annotation data improved the prediction accuracy by 2.0% and 3.8% when using 1 M HapMap3 and 7 M common SNPs,

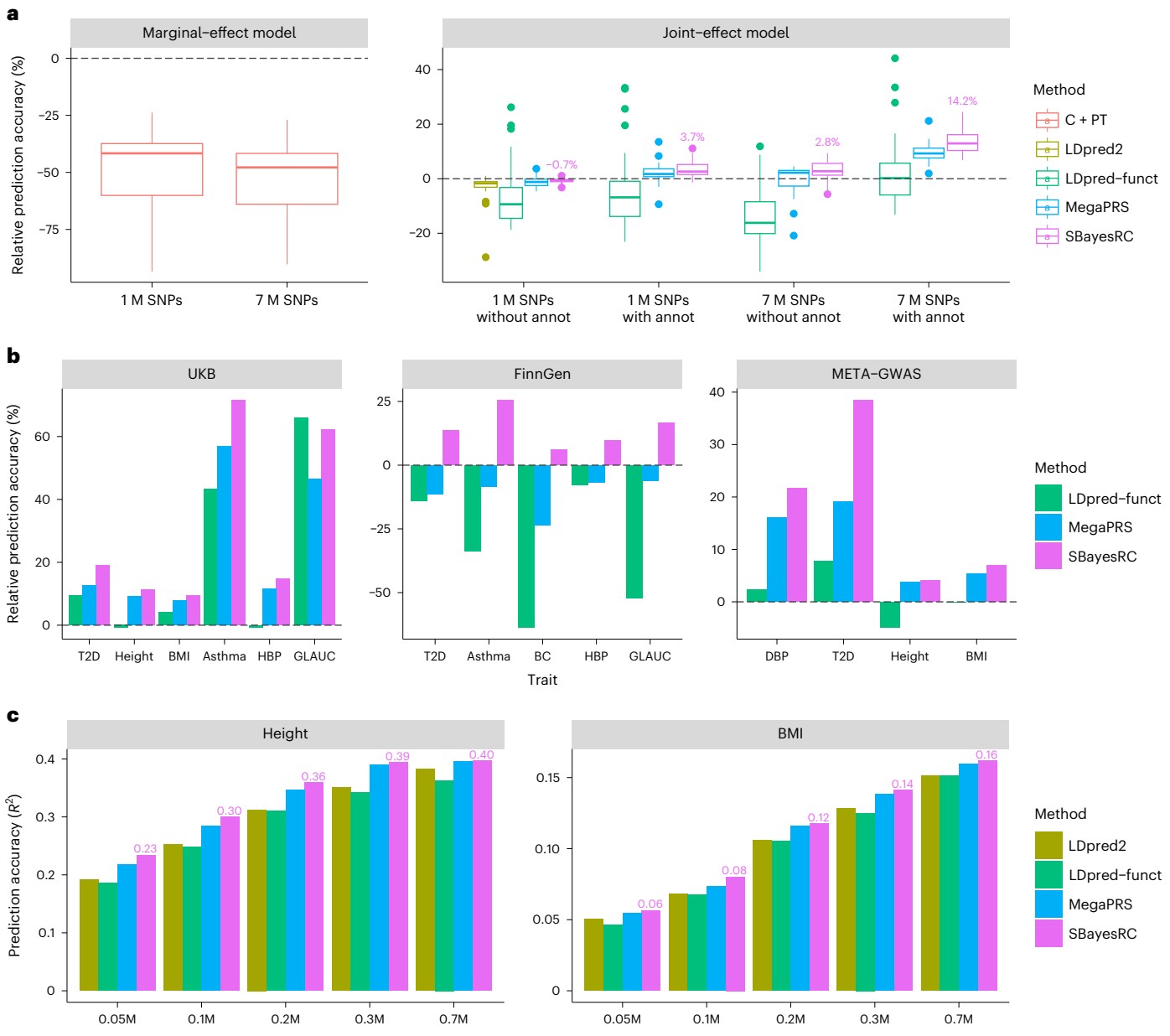

**Fig. 3 | Prediction performance using SBayesRC with 7 M SNPs and annotation data in European populations. a**, Relative prediction accuracy of different methods to SBayesR using 1 M HapMap3 SNPs, averaged from ten-fold cross-validation in the UKB ($n$ = 28 traits). Each box plot shows the spread of data; the line is the middle (median), the box covers the middle half (IQR), the whiskers extend to 1.5 times the IQR, and dots show outliers. **b**, Relative prediction accuracy of different methods to LDpred2 (grid of models) using 1 M HapMap3 SNPs for six traits in the UKB cross-validation (average value), five traits in the cross-biobank prediction analysis using the FinnGen data as training and the UKB data as validation, and four traits in the out-of-sample prediction analysis using the published meta-GWAS as training and the LifeLines data as validation. **c**, Out-of-sample prediction accuracy for height and BMI, using the UKB ($n$ = 0.05 to 0.3 M by downsampling) or the GIANT dataset[40] ($n$ = 0.7 M) as training and the LifeLines data as validation.

respectively. Although a similar pattern was observed in LDpred-funct and MegaPRS, SBayesRC consistently outperformed both methods in each scenario (Fig. 2c and Supplementary Fig. 5). We hypothesize that the advantage of exploiting annotations arises from both better identification of causal variants and better estimation of their effect sizes. This hypothesis is supported by the results that incorporating annotations in the model led to higher power and lower false discovery rate (FDR) for identifying the causal variants (Fig. 2d, e) and a stronger correlation in the estimated and true SNP effects (Fig. 2f). Coupled with the higher prediction accuracy, the SNP-based heritability estimation approached the true value in the simulation when more SNPs with annotation data were used (Extended Data Fig. 4). Moreover, we demonstrated through sensitivity analyses that SBayesRC is robust in

various circumstances, including a misspecification of mixture distribution scaling factors or the number of mixture components, and using an alternative data-generative model for simulation (Supplementary Figs. 9–11 and Section 12 of the Supplementary Note).

### Improved prediction accuracy within European ancestry
For the evaluation of prediction accuracy within European ancestry, we conducted ten-fold cross-validation in the 28 approximately independent traits from the UKB and cross-biobank prediction using data from the LifeLines cohort[37] and the FinnGen project[38] (Methods and Supplementary Table 1). We used 96 genomic annotations from BaselineLD v2.2 (ref. 24) and 7 M imputed common SNPs in the UKB after matching with validation and annotation datasets (Methods).

To assess the performance of our method in comparison to different approaches, we considered the analysis of 1 M HapMap3 SNPs without any annotation using SBayesR[34] as the benchmark and ran other methods, including C + PT[39], LDpred2 (ref. 36), LDpred-funct[28] and MegaPRS[29]. The prediction accuracy of each method was assessed by calculating the relative value to that of SBayesR ($\delta_x^2 = \frac{R_x^2 - R_{\text{SBayesR}}^2}{R_{\text{SBayesR}}^2}$, where $R_x^2$ is the prediction R-squared of method $x$ in the validation sample). When using HapMap3 SNPs only, SBayesRC without annotations gave a prediction accuracy similar to that of SBayesR, which is significantly higher than that of LDpred2 ($\delta_{\text{LDpred2}}^2 = -3.2\%$, Wilcoxon signed rank exact test $P = 1.4 \times 10^{-7}$) (Fig. 3a). The use of 7 M SNPs or annotation data in SBayesRC resulted in an improvement in prediction accuracy by 2.8% ($P = 0.001$) or 3.2% ($P = 3.2 \times 10^{-7}$), respectively, on average across traits. The combined use of both sources of information further increased the prediction accuracy by 14.2% ($P = 7.5 \times 10^{-9}$), indicating a strong interaction between the SNP density and annotations (see more discussion below). MegaPRS exhibited the second highest mean prediction accuracy and a similar boost with the combination of 7 M SNPs and annotation information, comparable to the results from SBayesRC. Overall, SBayesRC outperformed LDpred-funct by 11.9% ($P = 5.5 \times 10^{-5}$) and MegaPRS by 4.1% ($P = 2.5 \times 10^{-7}$) in prediction accuracy, when using 7 M SNPs and annotation data. In addition, the regression slopes from SBayesRC were close to one across different traits, indicating that the SBayesRC predictors were unbiased (Extended Data Fig. 5). Consistent results were observed in an extended analysis of 50 complex traits (Extended Data Fig. 6 and Supplementary Tables 3 and 8).

We conducted two sets of cross-biobank prediction analyses using FinnGen and LifeLines datasets (Methods). In both cases, SBayesRC yielded the highest prediction accuracy, consistent with the results from the UKB cross-validation (Fig. 3b). Particularly, SBayesRC demonstrated significant advantages in the analysis of FinnGen summary statistics, whereas both MegaPRS and LDprep-funct had lower prediction accuracy than LDpred2, which only used 1 M HapMap3 SNPs without annotations. The significant advantage of SBayesRC over MegaPRS can be attributed to its ability to better account for LD differences between GWAS and reference samples, which is further supported by the results of cross-biobank prediction within other ancestries (Extended Data Fig. 7). To explore the influence of sample size on prediction accuracy, we focused on height and body mass index (BMI) for which publicly available GWAS summary statistics with varying sample sizes were used. As expected, the prediction accuracy improved with increasing training sample size for both height and BMI in all methods. SBayesRC consistently outperformed LDpred2 by 4.0–21.9% and LDpred-funct by 7.1–26.3% and performed slightly better than MegaPRS in each sample size (Fig. 3c). In the largest sample size analyzed ($n_{\text{GIANT}} = 0.7M^{40}$) and using SBayesRC with 7 M SNPs and 96 per-SNP annotations, we achieved a maximum prediction $R^2$ of 0.40 for height and 0.16 for BMI in the LifeLines cohort.

## Improved accuracy in cross-ancestry prediction

To assess whether the improved accuracy achieved by using functional annotations with genome coverage for prediction is transferable to populations of different ancestries, we performed cross-ancestry prediction in the UKB, where we trained predictors based on GWAS data from individuals of European (EUR) ancestry and validated in samples of South Asian (SAS), East Asian (EAS) and African (AFR) ancestries (Methods).

We evaluated SBayesRC, MegaPRS and two recently developed methods designed specifically for cross-ancestry prediction, PolyPred-S[15] and PRS-CSx[14] (Extended Data Table 1 lists a summary of these methods). PolyPred-S incorporates functional annotations through a fine-mapping analysis, whereas PRS-CSx combines information from multiple GWAS datasets, both requiring a tuning sample of individual-level data from the target population to generate the final SNP weights for prediction. We also allowed SBayesRC and MegaPRS to utilize these extra datasets by first running the method in each of the GWAS datasets of different ancestries separately and then combining the SNP effects with weights estimated from the tuning data (referred to as SBayesRC-multi and MegaPRS-multi; Methods).

In cross-ancestry prediction, we observed a decrease in prediction accuracy relative to that within EUR (Fig. 4a), which is consistent with previous studies[14–19,41]. However, despite the overall decline in prediction accuracy, the use of high-density SNPs beyond HapMap3 or functional annotation data led to increased prediction accuracy when compared to the benchmark of SBayesR within each of the ancestries (Fig. 4b). Within all non-EUR populations, SBayesRC using both 7 M SNPs and annotation data consistently achieved the highest prediction accuracy, with a relative improvement of 16.0% in SAS ($P = 1.5 \times 10^{-5}$), 22.6% in EAS ($P = 2.1 \times 10^{-4}$) and 33.7% in AFR ($P = 4.6 \times 10^{-5}$), averaged across traits. On average across the three non-EUR ancestries, SBayesRC outperformed PolyPred-S by 15.4% in mean prediction accuracy. MegaPRS with 7 M SNPs outperformed its 1 M SNPs counterpart and exhibited comparable prediction accuracy to SBayesRC (slightly worse by 3.3% on average across ancestries), but with a larger variance across traits. When using an additional set of GWAS summary statistics from Biobank Japan[42] (BBJ), PRS-CSx showed a 17.4% improvement compared to the benchmark of SBayesR in predicting EAS individuals in the UKB, slightly higher than that of SBayesRC using EUR data only but with annotations (15.9%). However, when SBayesRC-multi was used, which combines 7 M SNPs, functional annotations and the BBJ data, the improvement was almost doubled (32.9%), outperforming PRS-CSx by 13.5% (Fig. 4c). Similar patterns of improvement from the use of high-density SNPs and annotation data were observed in prediction within the AFR ancestry in PAGE[43] dataset (Fig. 4d). Notably, SBayesRC using EUR dataset only has readily outperformed PRS-CSx using both EUR and AFR datasets. Through combining all sources of information, SBayesRC-multi outperformed PRS-CSx and MegaPRS-multi by 40.9% and 7.7 % in mean prediction accuracy, respectively, and had smaller variance. These results demonstrate that leveraging functional annotations with all imputed SNPs can be as or more advantageous than using multiple GWAS datasets at a subset of SNPs, highlighting the importance of incorporating both types of information for optimizing cross-ancestry prediction.

In addition to improved prediction accuracy, SBayesRC also demonstrated efficient use of computational resources compared to

**Fig. 4 | Cross-ancestry prediction using SBayesRC with 7 M SNPs and annotation data. a**, The ratio of prediction accuracy for SBayesRC (with different SNP densities and whether using annotations), MegaPRS and PolyPred-S in each ancestry to that of SBayesR with 1 M HapMap3 SNPs averaged across ten folds of cross-validation in European ancestry ($n = 17$ traits). **b**, Relative prediction accuracy (% of improvement) of each method to that of SBayesR trained in the GWAS of European ancestry and validated in each of the other ancestries ($n = 17$ traits). VitD in PolyPred-S AFR population had a value of 331%, which is removed from the graph for a clear presentation. **c**, Relative prediction accuracy (as in **b**) either using summary statistics from UKB of European ancestry alone or together with those from BBJ of East Asian ancestry for cross-ancestry prediction in the UKB population of East Asian ancestry ($n = 8$ traits available). The number above each box plot indicates the mean value across traits. **d**, Relative prediction accuracy (similar to **c**) from UKB of European ancestry alone or together with those from PAGE of mixed African ancestry for cross-ancestry prediction in the UKB population of African ancestry ($n = 8$ traits available). White blood cell in **d** is an outlier in UKB EUR + PAGE (relative improvement of 140.7%, 173.3%, 188.8%, 189.3%, 211.6% and 128.2% for each method/scenario), which is removed from the plot for a clear presentation. Each box plot shows the spread of data; the line is the middle (median), the box covers the middle half (IQR), the whiskers extend to 1.5 times the IQR, and dots show outliers. Data are provided in Supplementary Tables 4–6.

other methods (Table 1). For the analysis of 7 M SNPs with 96 annotations, SBayesRC required 74 GB RAM and 8.5 computing hours with 4 CPU cores, which are commonly available in a standard computing cluster.

## Significant interaction between SNP density and annotation information

Results above have shown that the combination of the full imputation SNP set and annotation data outperformed the use of either one alone,

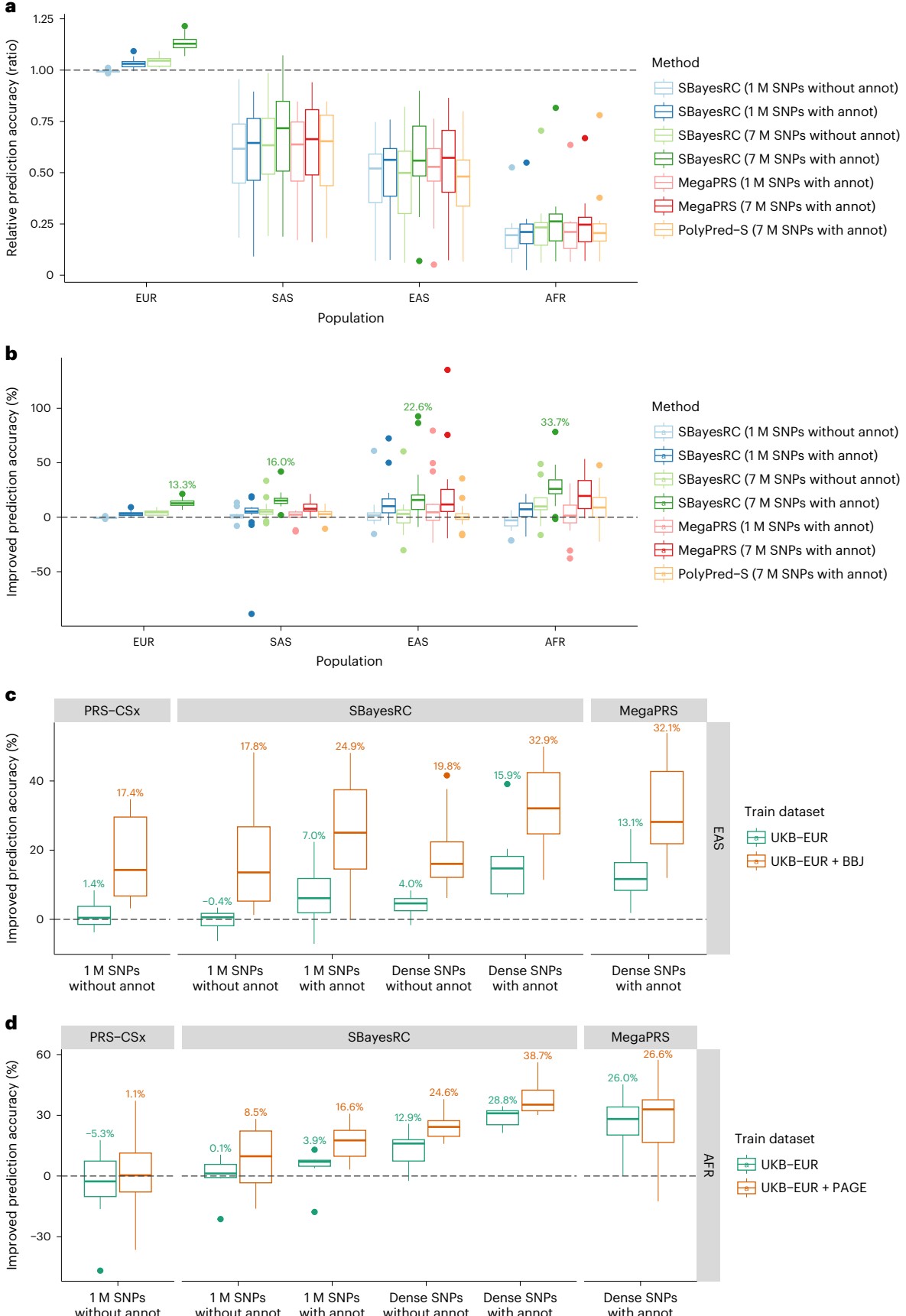

**Table 1 | Computation resource required for different methods[a]**

| Method (no. SNPs) | Runtime (h) | Memory (GB) | Required storage (GB) |
|---|---|---|---|
| SBayesRC (7M) | 8.5 | 73.3 | 72 |
| LDpred-funct (7M) | 6.0 | 120.6 | 40–50 per trait |
| PolyPred-S (7M) | 19.8[b] | 71.7 | 2,800 |
| MegaPRS (7M) | 7.2 | 247.7 | 277 |
| LDpred2 (1M) | 5.5 | 53.4 | 43 |
| SBayesRC (1M) | 0.8 | 4.8 | 5.6 |
| SBayesR (1M) | 0.5 | 27.0 | 22 |
| PRS-CSx (1M) | 14.2[c] | 4.7 | 5.6 |
| MegaPRS (1M) | 0.2 | 11.6 | 7.4 |

[a]Results were average values across traits using 4 CPU cores when multi-thread was supported for the method (*n* = 28 traits). Benchmarked with CPU AMD EPYC 7643 on a computing cluster. [b]In PolyPred-S, fine-mapping is the most time-consuming step and is suggested to run by blocks in parallel. Here we used a single core and divided the runtime by 4 for comparison to others. [c]Per dataset runtime: total runtime/number of training datasets (=2).

indicating an interaction effect between SNP density and annotation information. To investigate this interaction, we quantified the improvement in prediction accuracy due to the use of annotation data at each SNP density level (Methods). In the 28 independent UKB traits in EUR, the relative prediction accuracy with annotations versus without annotations at 7 M imputed SNPs was significantly greater than that at 1 M HapMap3 SNPs, with a twofold difference or more in most traits (Fig. 5). This difference was also observed in the cross-ancestry prediction, although with some variation (Extended Data Fig. 8). We performed a statistical test on this interaction by fitting the indicator variables for SNP density and annotation data, as well as their product, to the scaled prediction accuracy for each trait in UKB EUR (Methods). The test showed that the interaction effect was highly significant ($P_{Interaction}$ = 6.7×10$^{-7}$), in addition to significant main effects for SNP density ($P_{SNP density}$ = 4.2 × 10$^{-4}$) and annotations ($P_{Annotations}$ = 1.1 × 10$^{-5}$). Similar significant interaction effect was also observed in MegaPRS ($P_{Interaction}$ = 1.1 × 10$^{-5}$) and LDpred-funct ($P_{Interaction}$ = 0.048) (Fig. 3a), suggesting that this phenomenon is capturing a biological signal independent of the prediction methods. This finding is in line with the hypothesis that the annotations at the SNPs in LD with a causal variant may not accurately reflect the annotation at the causal variant itself, resulting in a loss of information (Fig. 1a).

### Other factors affecting accuracy of prediction leveraging functional annotations

Here, we investigate other factors, besides SNP density, that affect accuracy of prediction leveraging functional annotations, including SNP-based heritability, GWAS sample size, properties of minor allele frequency (MAF) and LD, the number of annotations and the analysis strategy. The results showed that traits with lower SNP-based heritability or smaller GWAS sample sizes tended to benefit more from leveraging annotation data for prediction (Fig. 6a, b). Analyses focusing on height and BMI showed that functional annotations were more informative than LD and MAF annotations, and using a comprehensive set of functional annotations was superior to using only a few key functional categories (Fig. 6c). Moreover, we found that the unified analysis using all 7 M SNPs in the model was better than the stepwise analysis in refining the information from annotation data (Fig. 6d). Details of these analyses are described in Section 16 of the Supplementary Note.

### Contributions of functional categories to prediction accuracy

To identify which functional annotations are most important, we constructed functional category-specific PGS using SNPs within that

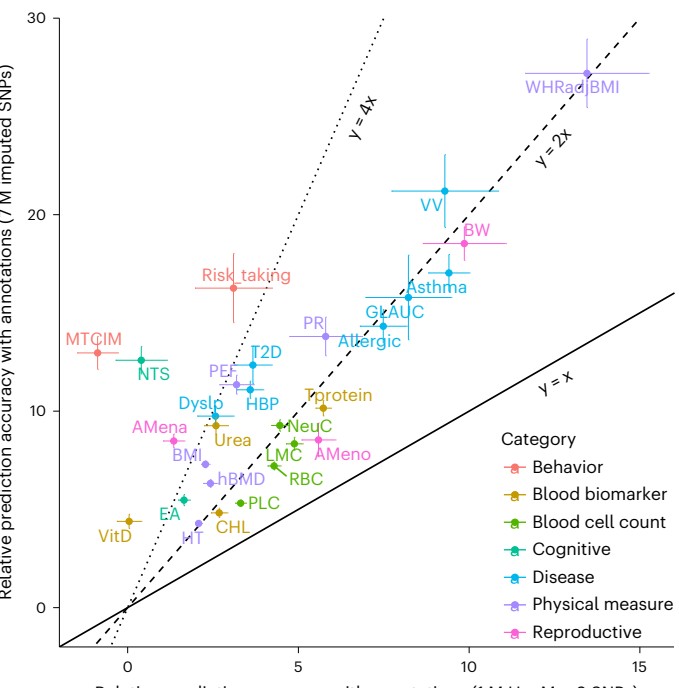

**Fig. 5 | Comparison between 1 M HapMap3 SNPs and 7 M imputed SNPs for the improvement in prediction accuracy for SBayesRC using annotations relative to SBayesRC without annotations.** Results are from the ten-fold cross-validation in the unrelated UKB samples of European ancestry. Dot shows the mean relative prediction accuracy, and bar shows the standard error estimated from the cross-validation. Color shows trait category; the definitions for the trait acronyms are provided in Supplementary Table 1.

functional category and their effect estimates from the genome-wide analysis of SBayesRC. Overall, categories with more SNPs made a greater contribution to the prediction accuracy, but there were some apparent outliers (Fig. 7a). Notably, evolutionary constrained regions, despite being small in SNP set size, had the greatest contribution among all categories without flanking windows. For example, regions that are conserved across 29 eutherian mammals (Conserved_LindbladToh[44] in BaselineLD) only cover 2.9% of the genome but contributed 40.5% of the prediction accuracy averaged across traits, resulting in a per-SNP predictability enrichment of 14.0-fold (that is, enrichment in per-SNP contribution to prediction accuracy = 40.5/2.9). In comparison, the coding regions (which account for 1.6% of the genome) contributed 25.9% of the prediction accuracy, with a per-SNP predictability enrichment fold of 16.5. This result suggests that evolutionary constrained variants are as informative as the coding variants for complex trait prediction. Across functional categories, the per-SNP contribution to prediction accuracy was proportional to the per-SNP contribution to heritability (Fig. 7b), suggesting that the variance explained by an SNP in the GWAS sample can be transferred into its predictive ability in the validation sample. Nonsynonymous SNPs in the coding sequence showed the largest per-SNP predictability (41.4-fold enrichment), and they also exhibited the largest enrichment in per-SNP heritability.

We prioritized functional annotations based on their per-SNP heritability enrichment, averaged across the traits analyzed in this study. The top 20 annotations showed a mean fold enrichment in per-SNP heritability ranging from 3.8 to 18.8, which included nonsynonymous variants, evolutionary constrained regions, coding sequence and regulatory elements. These results were by and large consistent with the results from S-LDSC (Supplementary Fig. 12). Notably, our method allows us to go on ask whether the enrichment in per-SNP heritability was due to a higher number of causal variants or larger effect sizes in the category. We found that, conditional on the other annotations, the

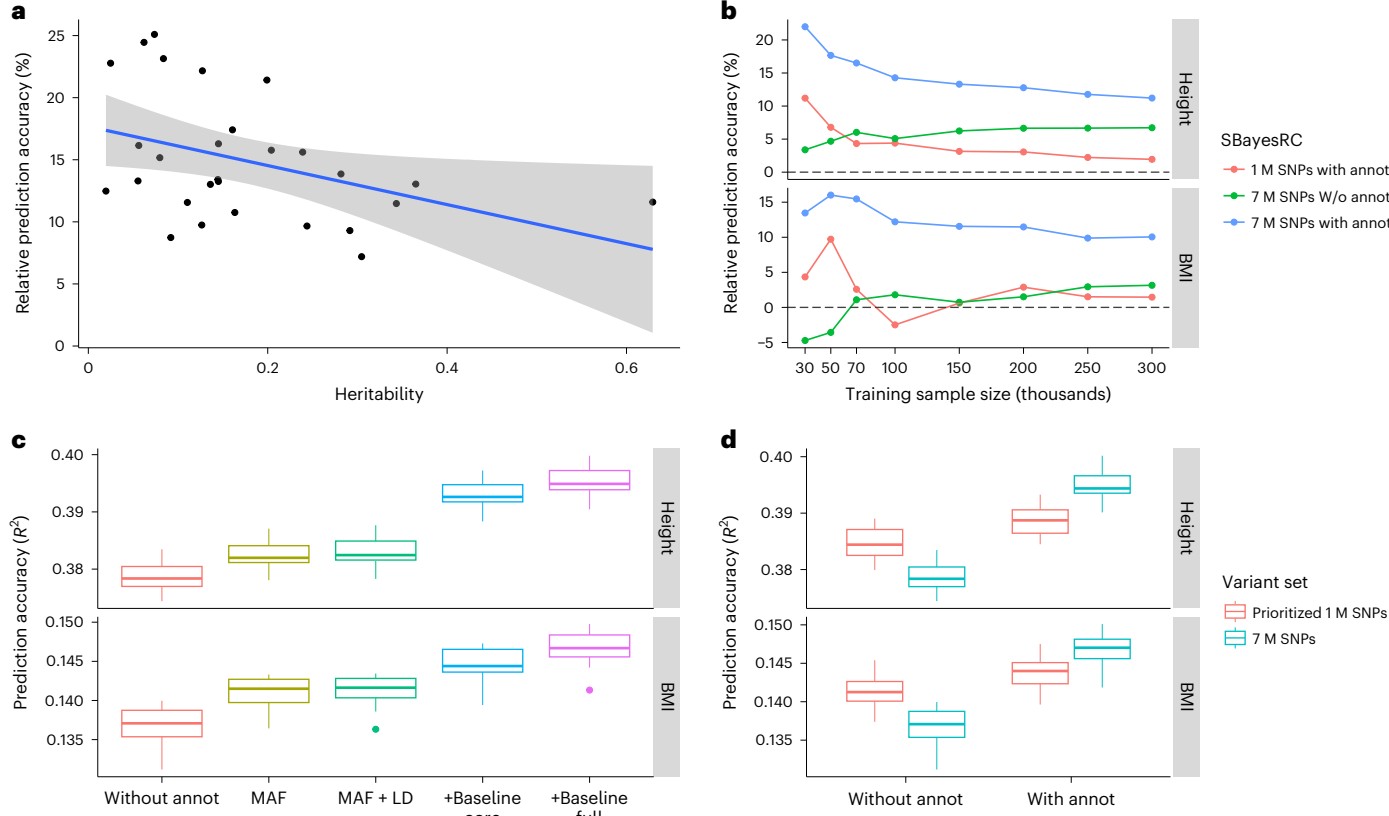

**Fig. 6 | Other factors affecting accuracy of prediction incorporating functional annotations. a**, Traits with low heritability tend to benefit more from using annotation data. The blue solid line indicates the linear regression of the data points and shading indicates the confident interval of the regression. **b**, GWASs with small sample sizes tend to benefit more from using annotation data. **c**, Improvement in prediction accuracy increases with the number of

annotations upon the MAF and LD (+Baseline core/full = MAF + LD+Baseline core/full set of annotations). **d**, Full analysis of all SNPs and annotation data is superior to the stepwise analysis that prioritizes the top 1 M SNPs based on their annotations and fits them in the model. Each box plot in **c** and **d** shows the spread of data in ten cross-validations; the line is the middle (median), the box covers the middle half (IQR), the whiskers extend to 1.5 times the IQR and dots show outliers.

nonsynonymous SNPs category was enriched in both the proportion of causal variants and the magnitude of effect sizes (Fig. 7c). Moreover, compared to evolutionary conserved regions in mammals, conserved regions in primates had lower proportion of null SNPs and higher proportions of SNPs with small to large effects in human traits.

## Discussion

We have introduced a novel method, SBayesRC, for polygenic prediction of complex traits using GWAS summary statistics of the full set of imputed SNPs and incorporating diverse functional annotations on each SNP. Compared to the common practice of using 1 M HapMap3 SNPs, leveraging 7 M imputed common SNPs and 96 per-SNP annotations resulted in a 14% improvement in prediction accuracy within European ancestry across 28 complex traits and diseases, and up to 34% improvement across ancestries averaged over 18 well-powered traits. These results indicate that incorporating functional annotations into prediction models can significantly enhance prediction accuracy, consistent with previous studies[15,26–28,30,45]. SBayesRC outperformed the best methods for both within European ancestry and cross-ancestry prediction using annotations, MegaPRS and PolyPred-S, suggesting its superiority in leveraging annotation data for prediction. SBayesRC-multi outperformed PRS-CSx, highlighting the importance of considering both annotation data and multiple GWAS datasets for cross-ancestry prediction. Furthermore, this study revealed a significant interaction between SNP density and annotation information for prediction accuracy, indicating that the benefits of incorporating annotations into prediction are amplified with higher SNP density.

The interaction between SNP density and annotation information can be explained as follows. First, when using a low-density panel of SNPs, the available information from functional annotations may not provide an accurate prior for weighing the SNP effects, because the SNPs in the low-density panel may not be the causal variants but instead may be in LD with them. In this case, SNPs carrying different annotations could capture the effects of the causal variants, resulting in a misspecification of the SNP effect prior and potentially biased estimation of annotation effects (Fig. 1a). Indeed, as shown by simulation, the estimation of the proportion of SNPs in each non-zero distribution was unbiased in the full SNP panel but significantly biased in the 1 M subset of SNPs (Supplementary Fig. 7). Second, using a high-density panel of SNPs allows for better fine-mapping of the causal variants and better estimation of their effects. Consistent with the previous studies[46,47], the additional information from the annotation data effectively enhanced the power in fine-mapping causal variants (Fig. 2d,e) and the accuracy of estimating causal effects (Fig. 2f). In the real data analysis, a significant improvement was also observed from 1 M to 7 M imputed SNPs with annotations, but no further difference was observed with 10 M imputed SNPs (Extended Data Fig. 9). This plateau in prediction performance could be attributed to the saturation of SNP tagging on the common causal variants by the 7 M set or due to limitations in imputation accuracy on common SNPs or sample size of discovery GWAS.

We found that the combination of high-density SNPs and functional annotations provides the most benefit to traits with low SNP-based heritability or small GWAS discovery sample sizes by providing additional information to allele frequency and LD categories.

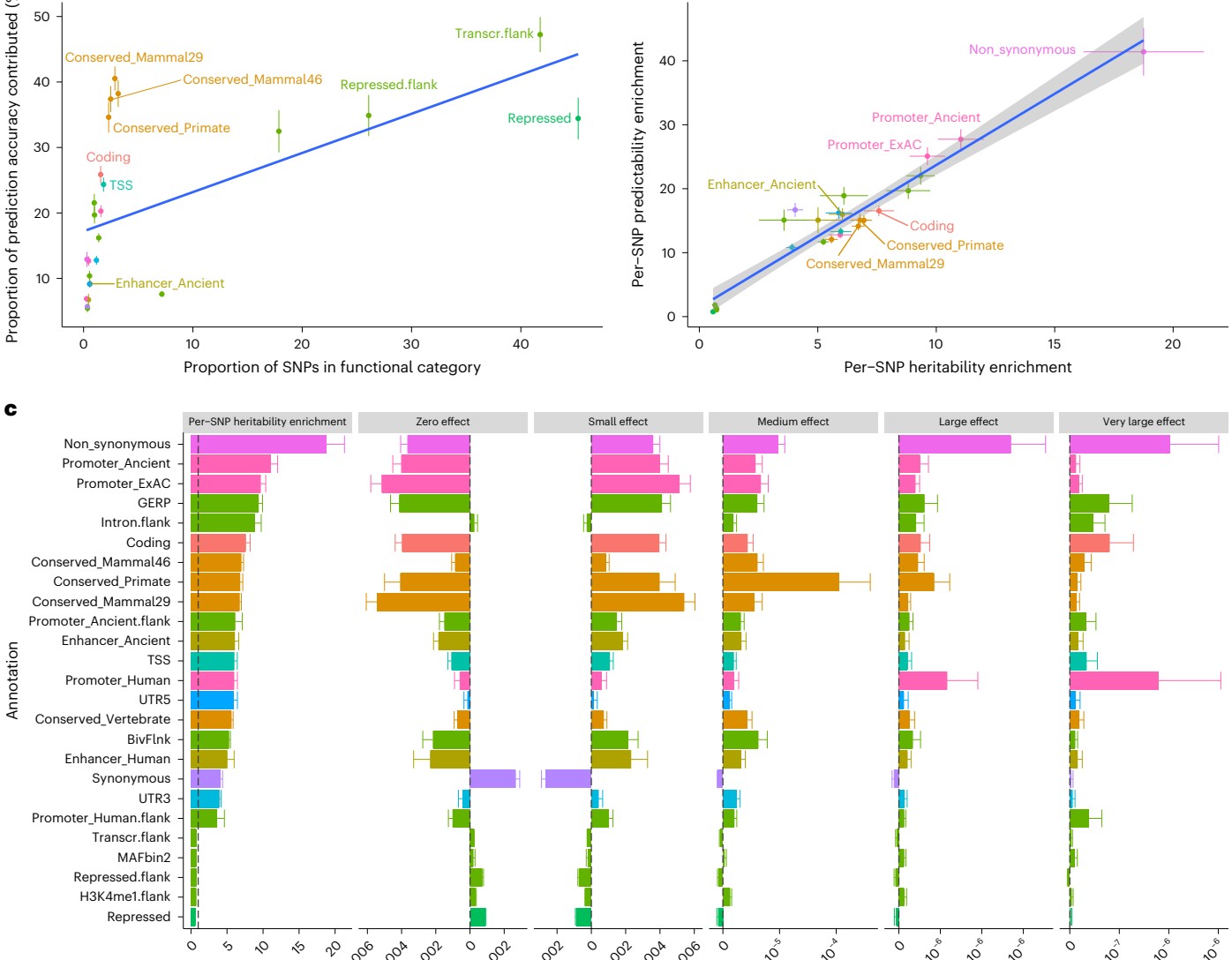

**Fig. 7 | Contribution of functional categories to the total prediction accuracy and estimation of functional genetic architecture in complex traits. a,** Proportion of prediction accuracy against proportion of SNPs in each functional category. **b,** Per-SNP contribution to prediction accuracy against per-SNP contribution to heritability in each functional category. The dots in **a** and **b** show the mean value from 28 traits in one functional category, error bar shows the standard error, the blue solid line indicates the linear regression of the data points and shading in **b** indicates the confident interval of the regression.

**c,** Per-SNP heritability enrichment and distribution of effect sizes for the top 20 and bottom 5 functional categories. The distribution of effect sizes for each functional category is shown as the deviates of the proportion of SNP effects belonging to each of the five mixture distributions to the overall proportion of genome-wide SNPs across functional categories. Each bar plot in **c** shows the mean value from 28 traits in one functional category, and error bar shows the standard error. The mapping to original categories in BaseLineLD model v2.2 and the data are provided in Supplementary Table 9.

These results highlight the utility of leveraging functional annotations for predicting disease risk, as most common diseases do not have a high SNP-based heritability and the effective sample sizes are still limited for many diseases. Additionally, our findings underscore the importance of generating more high-quality functional annotations, as they offer biological information beyond non-functional dependent annotations like MAF and LD. Furthermore, we demonstrated that using a unified computational framework to jointly model the GWAS and annotation data is more desirable than the stepwise approaches commonly used in the previous studies[28,48]. The results of this study are useful to inform the experimental design of leveraging functional annotations for prediction in future research.

We note several limitations in this study. First, although our method is scalable to analysis of whole-genome sequence data, we only analyzed imputed common SNPs that were functionally annotated due to limitations in the availability of whole-genome sequence data during the study. We investigated use of up to 10 million imputed SNPs with MAF > 0.01 but did not observe a significant improvement comparing to the 7 million SNP set. A follow-up study with sequence variants is warranted to explore this further. Second, our low-rank model requires the GWAS summary data to match the SNPs used to generate the LD data; otherwise, eigen-decomposition on the LD matrices would need to be recomputed. An alternative approach is to impute the summary statistics for those 'missing' SNPs[49] (Section 17 of the Supplementary Note). We found empirically that the loss of prediction accuracy was marginal unless the missing rate exceeded 30%. Third, although our method has improved robustness to LD and per-SNP sample size variation, it is still subject to other errors in the GWAS summary statistics,

such as genotyping errors and allelic mislabeling. Thus, application of additional quality control on the summary statistics prior to the analysis may be necessary in some circumstances[50]. Fourth, for cross-ancestry prediction, there is possibility of further improvement in prediction accuracy by jointly modeling summary statistics from multiple populations, as done in PRS-CSx[14]. However, we leave such an extension of our method to a future project. Fifth, this study used general annotations curated by the BaselineLD model[32], which does not include annotations from recent studies regarding cell-type specific epigenetic marks and chromatin states[51–54]. Incorporating annotations derived from the trait-relevant tissues or cell types, as inferred from GWAS data and single-cell omics data, is expected to generate more accurate predictors. As irrelevant annotations may slightly decrease in prediction accuracy when the GWAS power is relatively low (Supplementary Fig. 8), we recommend utilizing biologically informative annotations, particularly for traits with limited power.

In conclusion, the method proposed in this study is a powerful approach to improve polygenic prediction in complex traits and diseases. Our findings provide guidelines on how to best utilize functional annotation data for prediction and which functional categories are most useful for within European and cross-ancestry prediction. We anticipate further improved prediction accuracy in the future when the method is applied to whole-genome sequence data with high-quality trait-relevant annotations.

## Online content

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

[1]Institute for Molecular Bioscience, The University of Queensland, Brisbane, Queensland, Australia. [2]Program in Medical and Population Genetics, Broad Institute of Harvard and MIT, Cambridge, MA, USA. [3]Stanley Center for Psychiatric Research, Broad Institute of Harvard and MIT, Cambridge, MA, USA. [4]Center for Economic and Social Research, University of Southern California, Los Angeles, CA, USA. [5]Department of Economics, University of Southern California, Los Angeles, CA, USA. [6]Department of Epidemiology, University of Groningen, University Medical Center Groningen, Groningen, the Netherlands. [7]Department of Bioinformatics, Isfahan University of Medical Sciences, Isfahan, Iran. [8]School of Life Sciences, Westlake University, Hangzhou, Zhejiang, China. [9]Westlake Laboratory of Life Sciences and Biomedicine, Hangzhou, Zhejiang, China. [10]Department of Psychiatry, University of Oxford, Oxford, UK. [11]Faculty of Veterinary and Agricultural Science, University of Melbourne, Parkville, Victoria, Australia. [12]Biosciences Research Division, Department of Economic Development, Jobs, Transport and Resources, Bundoora, Victoria, Australia. [13]Big Data Institute, Li Ka Shing Centre for Health Information and Discovery, Nuffield Department of Population Health, University of Oxford, Oxford, UK. ✉e-mail: zhili.zheng@broadinstitute.org; j.zeng@uq.edu.au

**LifeLines Cohort Study**

**Raul Aguirre-Gamboa[14], Patrick Deelen[14], Lude Franke[14], Jan A. Kuivenhoven[15], Esteban A. Lopera Maya[14], Ilja M. Nolte[6], Serena Sanna[14], Harold Snieder[6], Morris A. Swertz[14], Peter M. Visscher[1,13], Judith M. Vonk[6] & Cisca Wijmenga[14]**

[14]Department of Genetics, University of Groningen, University Medical Center Groningen, Groningen, the Netherlands. [15]Department of Pediatrics, University of Groningen, University Medical Center Groningen, Groningen, the Netherlands.

## Methods

### Ethics approval

The University of Queensland Human Research Ethics Committee B (2011001173) provides approval for analysis of human genetic data used in this study on the high-performance cluster of the University of Queensland.

### Summary-data-based low-rank model

Consider a general form of the summary-data-based model for fitting SNP joint effects:

$$\mathbf{b} = \mathbf{R}\boldsymbol{\beta} + \boldsymbol{\varepsilon} \qquad (1)$$

where $\mathbf{b}$ is the vector of GWAS marginal effect estimates (assuming the genotype matrix $\mathbf{X}$ has already been standardized with mean zero and variance one), $\mathbf{R} = \frac{1}{N}\mathbf{X}'\mathbf{X}$ is the LD correlation matrix, $N$ is the GWAS sample size, $\boldsymbol{\beta}$ is the vector of SNP joint effects, and $\boldsymbol{\varepsilon}$ is the vector of residual terms with $Var(\boldsymbol{\varepsilon}) = \frac{1}{N}\mathbf{R}\sigma_e^2$. When the marginal effects are estimated from GWAS using genotypes at 0/1/2 scale ($\mathbf{b}^*$), $\mathbf{b}$ can be estimated using $\mathbf{b}^*$, standard error and GWAS sample size (Section 6 of the Supplementary Note).

Sparse LD matrices estimated from a reference sample are often used to improve computational feasibility, including banded[28,36], shrunk[34,55] and block-diagonal[14,56] matrices. For our low-rank model, we use a block-diagonal LD matrix based on quasi-independent LD blocks found in the human genome[35]. For optimal performance, we merge small contiguous blocks into a single block with the minimum width of 4 cM, resulting in 591 merged blocks for the samples of European ancestry. For each block $i$, we perform eigen-decomposition on $\mathbf{R}_i$ (the subscript is ignored for simplicity in notation)

$$\mathbf{R} = \mathbf{U}\boldsymbol{\Lambda}\mathbf{U}',$$

where $\mathbf{U}$ is the matrix of eigenvectors and $\boldsymbol{\Lambda}$ is the diagonal matrix of eigenvalues. By multiplying both sides of Equation 1 by $\boldsymbol{\Lambda}^{-\frac{1}{2}}\mathbf{U}'$, we have

$$\mathbf{w} = \mathbf{Q}\boldsymbol{\beta} + \boldsymbol{\epsilon} \qquad (2)$$

where $\mathbf{w} = \boldsymbol{\Lambda}^{-\frac{1}{2}}\mathbf{U}'\mathbf{b}$ is a linear combination of marginal SNP effect estimates, $\mathbf{Q} = \boldsymbol{\Lambda}^{\frac{1}{2}}\mathbf{U}'$ is the new coefficient matrix and the new residuals $\boldsymbol{\epsilon} = \frac{1}{N}\boldsymbol{\Lambda}^{-\frac{1}{2}}\mathbf{U}'\mathbf{X}'\mathbf{e}$ are independently and identically distributed, that is, $\boldsymbol{\epsilon} \sim N(\mathbf{0}, \mathbf{I}\sigma_\epsilon^2)$, making it straightforward to estimate the residual variance, thereby improving the model robustness (Section 3 of the Supplementary Note). To account for high LD between SNPs and LD variations between GWAS and LD reference samples, we opt to include eigenvectors and eigenvalues for the top principal components (PCs) that collectively explain at least $\rho$ proportion of the variance in LD. Assuming $q$ top PCs are selected given a value of $\rho$, the dimension of $\mathbf{w}$ and $\mathbf{Q}$ is $q \times 1$ and $q \times m$, respectively, with $m$ being the number of SNPs in the block. Because $q$ is often much smaller than $m$, Equation 2 is a low-rank model and computationally more efficient than Eq. (1). We investigated the impact of $\rho$ on the method and decided to use $\rho = 99.5\%$ as the default value with negligible loss in predictive performance (Supplementary Figs. 2 and 3; Section 9 of the Supplementary Note). However, the optimal value of $\rho$ in real trait analysis would depend on the LD variation between GWAS and reference datasets. To enable an automated search for the best $\rho$ for the trait, we performed pseudo validation based on the observed summary statistics, similar to the method used in Zhang et al.[29], but requires the result of eigen-decomposition of LD matrix that has already been generated (Section 10 of the Supplementary Note).

### SBayesRC

SBayesRC is a Bayesian method built on the low-rank model described above, assuming a multi-normal mixture distribution for SNP effects. Specifically, we assume

$$\beta_j \sim \sum_{k=1}^{5} \pi_{jk} N(0, \gamma_k \sigma_g^2),$$

where $\sigma_g^2$ is the total SNP-based genetic variance estimated from the data and $\boldsymbol{\gamma} = [0, 0.001, 0.01, 0.1, 1]'\%$ depict the scaling factors of five distributions as the mixture components, including a distribution of zeros and four normal distributions, where each SNP *a priori* explains 0.001% to 1% of genetic variance. The parameter $\pi_{jk}$ is the probability for the SNP effect to belong to the $k^{th}$ distribution.

In contrast to SBayesR[34], which assumes the same $\pi_k$ for all SNPs, here the probability of distribution membership $\pi_{jk}$ is SNP-specific and depends on the annotations of each SNP. Let $\mathbf{A}$ be the matrix of annotations with a dimension of the number of SNPs $m$ by the number of annotations $c$. For each SNP, we model $\pi_{jk}$ as

$$f(\pi_{jk}) = \mu_k + \sum_{l=1}^{c} A_{jl}\alpha_{kl} \qquad (3)$$

where $f(\cdot)$ is a link function that maps the probability variable $\pi_{jk}$ to the real line, $\mu_k$ is the intercept capturing the overall proportion of SNPs belonging to the $k^{th}$ distribution in the genome, $A_{jl}$ is the value of annotation $l$ on SNP $j$ (0 or 1 for binary annotations or standardized value with mean 0 and variance 1 for quantitative annotations), and $\alpha_{kl}$ is the effect of annotation $l$ on the membership probability to the $k^{th}$ distribution. This generalized linear model allows functional annotations to affect the probability of an SNP being causal ($1 - \pi_{j1}$) and accommodates any distribution of the causal effect (by mixture of a finite number of normal distributions) given the cumulation of functional annotations, regardless of discrete or quantitative annotations, accounting for overlapping between annotations. Through estimation of $\alpha_{kl}$ from the data, this computational framework provides a machinery to make inference on the functional genetic architecture of the trait, because $f^{-1}(\alpha_{kl})$ quantifies the deviation of the $k^{th}$ distribution membership probability, driven by annotation $l$, to the baseline model where all annotation values equal to zero, conditional on the presence of the other annotations. The estimates of $\alpha_{1l}, \dots, \alpha_{5l}$ altogether provide a more detailed description about functional architecture than the per-SNP heritability enrichment estimate for an annotation category (Section 7 of the Supplementary Note and Supplementary Figs. 7 and 13). We assume a flat prior for $\mu_k$ and a normal prior for $\alpha_{kl} \sim N(0, \sigma_{\alpha_k}^2)$ with $\sigma_{\alpha_k}^2 \sim \chi^{-2}(\upsilon_\alpha, \tau_\alpha^2)$, where $\upsilon_\alpha = 4$ and $\tau_\alpha^2 = 1$.

For a mixture distribution of five components, there are $5 \times (c + 1)$ annotation parameters to estimate from the data (including the intercept). In addition, $\pi_{jk}$ is subject to a constraint that $\sum_{k=1}^{5} \pi_{jk} = 1$ for any SNP, which makes the sampling scheme for $\alpha_{kl}$ not straightforward. Although the Metropolis–Hastings algorithm can be used to sample all $\boldsymbol{\alpha}$ jointly to account for the dependence between elements of $\boldsymbol{\pi}_j$, finding the optimal tuning parameters for the proposal distribution could be difficult and specific to the trait. To remove the dependence between probability parameters, we used an alternative parameterization for modeling membership probabilities and annotation effects. Let $\delta_j$ be the indicator for the mixture component membership for SNP $j$:

$$\delta_j = k \text{ with probability } \pi_{jk}; \ k = 1 \text{ to } 5.$$

We define a conditional probability that the SNP effect belongs to the $k^{th}$ distribution given that it has passed the bar for the $(k-1)^{th}$ distribution as

$$p_{jk} = \Pr(\delta_j \geq k | \delta_j \geq k - 1) \text{ for } k \geq 2$$

such that $\pi_{j1} = 1 - p_{j2}$, $\pi_{j2} = (1 - p_{j3})p_{j2}$, $\pi_{j3} = (1 - p_{j4})p_{j3}p_{j2}$, $\pi_{j4} = (1 - p_{j5})p_{j4}p_{j3}p_{j2}$ and $\pi_{j5} = p_{j5}p_{j4}p_{j3}p_{j2}$. We then apply the generalized linear model, Equation 3, to link $p_{jk}$ with $\boldsymbol{\alpha}_k$. In this parameterization, all $p_{jk}$ are independent, which means that $\boldsymbol{\alpha}_k$ can be sampled in parallel in each Markov chain Monte Carlo (MCMC) iteration, and $\alpha_{kl}$ can be sampled from its full conditional distribution using Gibbs sampling algorithm when the probit link function is chosen, namely, $f(p_{jk}) = \Phi(p_{jk})$ where

$\Phi(\cdot)$ is the cumulative density function of the standard normal distribution. More details about the alternative parameterization and the MCMC sampling scheme are described in Section 8 of the Supplementary Note. In all SBayesRC analyses in this study, we ran MCMC for 3,000 iterations with the first 1,000 iterations as burn-in, and the rest were used for posterior inference. Running a longer chain did not change the prediction accuracy in the simulation and real trait analysis.

## UKB

The UK Biobank (UKB) is a large volunteer cohort with sample size of more than 500,000 individuals from the United Kingdom[57]. It contains extensive phenotypic and genotypic information from the participants, and all participants signed informed consent with the protocol's approval from the National Research Ethics Service Committee. The genotype data was generated using two array chips, the Applied Biosystems UKB Axiom Array and the Applied Biosystems UK BiLEVE Axiom Array. SNP imputation was conducted by the UKB analysis team using reference panels from the Haplotype Reference Consortium[58] and the UK10K project[59]. We called the imputed data to BED format by PLINK[60] with best-guest calling, kept SNPs with MAF $\geq$ 0.01, Hardy-Weinberg equilibrium test $P \geq 10^{-10}$, and imputation info score $\geq$ 0.6 in the samples of European ancestry. We used the GCTA software[61] to remove the cryptic relatedness in the UKB based on the HapMap3 SNPs in each population (cutoff value of 0.05), keeping only unrelated samples. We further removed samples with mismatched sex information in phenotype and genotype, and samples that withdrew participation. The final dataset contained four ancestries: European (EUR, $n$ = 347,800), East Asian (EAS, $n$ = 2,252), South Asian (SAS, $n$ = 9,436) and African (AFR, $n$ = 7,006).

We matched the SNPs between UKB, the annotation baseline model BaselineLD v2.2 (ref. [62]) and the LifeLines cohort[37], resulting in 7,356,518 common SNPs and 1,154,522 HapMap3 SNPs. For a secondary analysis, we included up to 9,705,522 imputed common SNPs with their annotation data extracted from PolyPred-S[15], which used BaseLineLF (an extended version of BaseLineLD v2.2 to include annotations at the low-frequency variants). We randomly sampled 5,991 EUR samples as the tuning sample for C + PT[60] and LDpred2 (ref. [36]) and performed ten-fold cross-validation in the remaining samples ($n$ = 341,809). We extracted 53 traits with relatively large sample size ($n$ > 100,000) from all four ancestries. The phenotypes with continuous values were filtered within the range of mean $\pm$ 7 standard deviation (s.d.) and then rank-based inverse-normal transformed within each ancestry and sex group. To construct a set of independent traits, we pruned these 53 traits with pair-wise phenotypic correlation $|r|$ < 0.3, resulting in 31 independent traits for the prediction analysis, including 11 binary traits and 20 continuous traits. Three binary traits were further removed due to very low average prediction accuracy (mean $R^2$ < 0.01 among all methods in the European cross-validation). The final set of 50 traits included in this study, of which 28 were approximately independent, are shown in Supplementary Table 1.

## 1000 Genomes and UK10K data

In addition to the UKB, we used two other whole-genome sequence datasets for LD reference. We obtained genotype data from the 1000 Genomes Project (phase 3)[63] and kept samples labeled as 'GBR', 'CEU', 'TSI', 'IBS' and 'FIN' as samples of European ancestry. After extracting the same SNP set (7,356,518 SNPs) and removing the cryptic relatedness as above, we retained 494 unrelated samples. We also used the genotype data from the UK10K project[59], which consisted of 3,781 individuals and 45.5 million genetic variants. After extracting the same SNP set and conducting QC as above, we retained 3,642 unrelated samples.

## LifeLines cohort

From the LifeLines cohort[37], we used 36,305 samples and 17 million SNPs after imputation and QC (imputation info score > 0.3, MAF > 0.0001 and HWE > $10^{-6}$). We kept the samples with age >20 years old and removed the samples with the phenotypic value beyond the range of mean $\pm$ 5 s.d. for the quantitative traits analyzed in this study (height, BMI and diastolic blood pressure). We further removed the related samples and retained 11,842 unrelated samples for out-of-sample prediction. For type 2 diabetes, we had 179 cases in the retained sample.

## FinnGen data

We accessed publicly available summary statistics from FinnGen[38], which had a sample size of 342,499. We selected five traits (Supplementary Table 7) that had a large number of cases in both UKB and FinnGen (each >1,000) and similar trait definition. As the LD reference was not publicly available from the FinnGen, we used the LD reference from the UKB. We kept the SNPs common in both datasets and matched the alleles. We further removed SNPs with a difference in allele frequency between GWAS and LD reference larger than 0.2. The FinnGen data was used as the training dataset in the cross-biobank prediction analysis, where the validation dataset was the UKB sample of European ancestry.

## Public data from GWAS meta-analysis

We trained the prediction models using publicly available summary data from published GWAS meta-analysis for height[40] ($n$ = 704,823), body mass index (BMI)[40] ($n$ = 688,633), diastolic blood pressure (DBP)[64] ($n$ = 756,595) and type 2 diabetes (T2D)[65] ($n_{case}$ = 62,693). We kept the same variant set in the UKB and the LifeLines and extracted the SNPs with per-SNP sample size within mean $\pm$ 3 s.d. and a difference in allele frequency between GWAS and LD reference samples smaller than 0.2. The summary data were further processed using DENTIST[50] to filter the SNPs with potential errors, and SNPs with $P_{DENTIST}$ < 5 × $10^{-8}$ and $P_{GWAS}$ > 0.01 were removed. Finally, all the summary data were imputed to the same variant panel for further analysis. These summary statistics were used as the training dataset in the out-of-sample prediction analysis, where the validation dataset was the LifeLines cohort.

## Cross-validation in the UKB

We performed ten-fold cross-validation in the UKB with 341,809 unrelated individuals of European ancestry. We partitioned the total sample into ten equal-sized disjoint subsamples. In each fold, one subsample was retained as the validation set, whereas the other nine subsamples were used as training data. This process was repeated ten times. Summary statistics for each fold were generated by PLINK2 software[60] with sex, age and first 10 PCs as covariates. Linear regression was used for continuous traits, and logistic regression was used for binary traits. The cross-validation was performed for all independent traits using the following methods: clumping and $P$-value thresholding (C + PT) implemented in PLINK 1.9 software, SBayesR[34], SBayesRC, LDpred2 (ref. [36]), MegaPRS[29] and LDpred-funct[28]. For all methods, a random sample of 20,000 unrelated UKB individuals of European ancestry was used as the LD reference. For SBayesR and LDpred2, only 1 M HapMap3 common SNPs were used for the ease of computation. For C + PT, LDpred-funct, MegaPRS and SBayesRC, both 1 M and 7 M common SNP sets were used and incorporated 96 functional annotations from BaseLineLD model 2.2 (ref. [62]) when possible. The specific settings for each method are described in Section 18 of the Supplementary Note.

For each fold, PGS was calculated using genotypes from the independent validation set. The prediction $R^2$ was obtained from linear regression of phenotypes on the PGS for quantitative traits, and McFadden's pseudo-$R^2$ from logistic regression was used for binary traits. The final $R^2$ of PGS was calculated as the difference between the $R^2$ from the full model (PGS + sex + age + 10 PCs) and the null model (sex + age + 10 PCs). The relative prediction accuracy was then computed as $\frac{R_x^2 - R_{SBayesR}^2}{R_{SBayesR}^2}$,

where $x$ is the prediction method being compared, and $R^2$ is the prediction accuracy. The mean relative prediction accuracy was reported across ten folds. For binary traits, additional statistics such as the area under the receiver-operating characteristic curve and the odds ratio

per s.d. of PGS from the logistic regression conditional on sex, age and 10 PCs were computed (Supplementary Tables 3 and 8). Overall, the area under the receiver-operating characteristic curve and odds ratio statistics yielded consistent results with the pseudo-$R^2$ for measuring prediction accuracy in diseases.

## Cross-ancestry prediction

We performed two sets of cross-ancestry prediction analyses. In the first set of analyses, we used the summary statistics from all European (EUR) unrelated samples as the training data (sample sizes shown in Supplement Table 1). We excluded 500 tuning samples from each non-EUR ancestry for methods that require a tuning step, and these samples were not used in the PGS validation for all methods. We ran SBayesR, SBayesRC and MegaPRS using the summary statistics of UKB EUR, and then applied the estimated SNP effects directly to the genotypes of individual of SAS, EAS and AFR ancestries in the UKB. We ran PolyPred-S following its pipeline and optimized the SNP weights with tuning samples from the target population. In this analysis, we calculated two types of relative prediction accuracy for each trait. In the first type of relative prediction accuracy, we used the prediction accuracy of SBayesR with 1 M HapMap3 SNPs in EUR as the benchmark. In the second type of relative prediction accuracy, the benchmark was the prediction accuracy of SBayesR trained in EUR and validated in each of the other ancestries.

In the second set of prediction analyses, we used two sets of summary statistics, one from the UKB EUR and the other from a GWAS study with the same ancestry of the validation population. We ran PRS-CSx with GWAS summary statistics from the UKB EUR and from BBJ[42] or Population Architecture using Genomics and Epidemiology (PAGE)[43] datasets. Then we estimated the optimal weights to combine the two sets of PGS using the target tuning samples from the UKB. Following a similar strategy, we extended SBayesRC and MegaPRS to utilize GWAS data from multiple populations by running the method in each population separately and tuned the weights in the target population (SBayesRC-multi and MegaPRS-multi). The specific settings for different methods used in this analysis are described in Section 19 of the Supplementary Note.

## Detection of interaction between SNP density and annotation information

To investigate the interaction between SNP density and annotation information, we first quantified the improvement in prediction accuracy due to the use of annotation information by calculating the relative prediction accuracy from the full model that includes annotations to the basic model that excludes annotations $\left( \delta^2 = \frac{R^2_{Full} - R^2_{Basic}}{R^2_{Basic}} \right)$ at each SNP density level. Then, we evaluated the interaction between SNP density and annotation information by comparing $\delta^2$ between 7 M imputed ($\delta^2_{7M}$) and 1 M HapMap3 SNPs ($\delta^2_{1M}$). If the benefit of including annotations is independent of SNP density (that is, no interactive effect between SNP density and annotation information), $\delta^2_{7M}$ is expected to be equal to $\delta^2_{1M}$ (that is, equal amount of improvement in prediction accuracy regardless of whether 7 M or 1 M SNPs are used). To formally test this interaction effect, we fit the indicator variables for SNP density and annotation data, as well as their product (that is, interaction term), to the prediction accuracy for each trait. To account for the variability in prediction accuracy between traits because of trait heritability, the prediction $R^2$ from different scenarios (involving different SNP density levels and the use of annotations) for each trait was scaled relative to the prediction $R^2$ obtained using HapMap3 SNPs without annotations.

## Simulations

We performed two sets of simulations to assess the performance of our method. The first set of simulations was performed using 1 M HapMap3

SNPs for model calibration and robustness assessment. In this set, we randomly selected 10,000 variants from the whole genome as causal variants. Among these, 6,000 variants had small effects sampled from N(0, 0.01), 30 variants had medium effects sampled from N(0, 0.1), and 10 variants had large effects sampled from N(0, 1). To introduce unequal per-SNP sample sizes, we divided the training sample into two equal-sized cohorts and generated two sets of summary statistics. We then randomly sampled a proportion of SNPs to conduct a meta-analysis using the inverse variance method, simulating scenarios where only a subset of SNPs was in common between cohorts for the meta-analysis. The proportion of overlapping SNPs between the two cohorts was set to be 100%, 90%, 50% or 0%. In the second set of simulations, we used 7 M imputed SNPs and incorporated annotation data. The causal effects were sampled following the SBayesRC model, where we used the annotation effects estimated from height in real data analysis and calculated the per-SNP probability of membership in each mixture component by probit link function, and then sampled the SNP effect from that distribution. Following Gazal et al.[62], we used 21 major annotations from the BaselineLD model. To evaluate the impact of different data-generative models, we also simulated data under the model of S-LDSC[32] (or MegaPRS without LD weighting), where the variance of SNP effect distribution is a function of annotations and their heritability enrichment. The details of this alternative model are provided in Section 13 of the Supplementary Note. In all simulations, normally distributed residuals were added to the genetic values to give a trait heritability of either 0.1 or 0.5. We repeated each simulation scenario ten times, with ten sets of different causal variants.

## Reporting summary

Further information on research design is available in the Nature Portfolio Reporting Summary linked to this article.

## Data availability

The UKB data, UK10K and LifeLines are available through formal application to the UKB (http://www.ukbiobank.ac.uk), UK10K (https://www.uk10k.org/data_access.html) and LifeLines (https://www.lifelines.nl/researcher/how-to-apply). The summary data and PGS weights from SBayesRC for the 28 approximately independent UKB traits can be found at https://cnsgenomics.com/software/gctb/#Download. All the other datasets used in this study are available in the public domain, including the 1000 Genomes Project (https://www.internationalgenome.org/data/), FinnGen (https://www.finngen.fi/en/access_results; version R8), BBJ (https://biobankjp.org/en/) and PAGE (https://www.ebi.ac.uk/gwas/publications/31217584).

## Code availability

SBayesRC is implemented in a R package at https://github.com/zhilizheng/SBayesRC and a publicly available software GCTB at https://cnsgenomics.com/software/gctb/#Download. A combined source code (both R package and GCTB) is also available at https://doi.org/10.5281/zenodo.10416921.

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

## Acknowledgements

We thank D. Stetner and the Delta cluster maintenance team from the Institute for Molecular Bioscience in the University of Queensland for their support in this research. We thank J. A. Revez, V. Hivert and H. Wang for assistance in processing the phenotypes. We thank M. Robinson, Y. Wu and X. Hu for helpful discussions. This research was supported by the Australian National Health and Medical Research Council (1177268), the National Institute of Mental Health (1R01MH121545-01) and the National Institutes of Health/National Institute on Aging grant 5R00AG062787 (PI Turley). This study makes use of data from the UKB (project ID 12505), FinnGen Biobank, Biobank Japan, PAGE, LifeLines Biobank, UK10K and 1KGP. We acknowledge the participants and investigators of these projects. A full list of acknowledgements for these datasets can be found in Section 21 of the Supplementary Note.

## Author contributions

J.Z. conceived and supervised the study. J.Z., Z.Z. and M.E.G. developed the methods and algorithms. J.Z., Z.Z. and P.M.V. designed the experiment. Z.Z. conducted all analyses with the assistance or guidance from S.L., J.S., Y.W., T.L., L.Y., P.T., J.Y., N.R.W., M.E.G., P.M.V. and J.Z. Z.Z., J.Z. and S.L. developed the software and R package. A.A., R.W., I.M.N. and H.S. provided the LifeLines Cohort Study data. Z.Z. and J.Z. wrote the manuscript with the participation of all authors. All the authors approved the final version of the manuscript.

## Competing interests

The authors declare no competing interests.

## Additional information

**Extended data** is available for this paper at https://doi.org/10.1038/s41588-024-01704-y.

**Correspondence and requests for materials** should be addressed to Zhili Zheng or Jian Zeng.

**a**

### Low-rank model

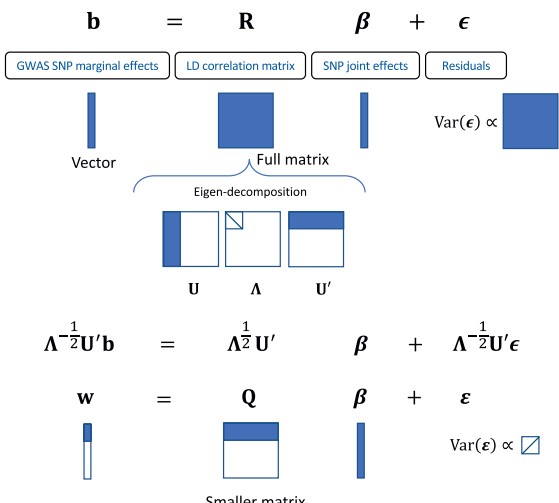

**b**

### Modelling functional annotation data

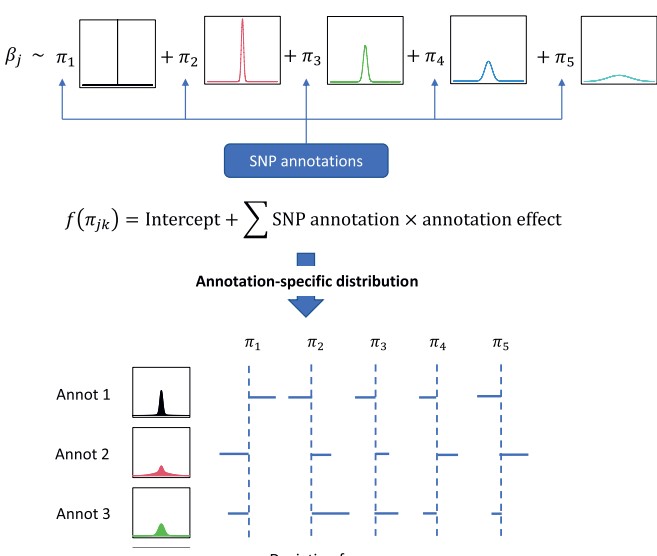

**Extended Data Fig. 1 | Schematic overview of SBayesRC. a**, A resource-efficient low-rank model that simultaneously fits sequence-level SNPs with high computation efficiency and has independent residuals. Solid blue box denotes a dense matrix or vector. **b**, A hierarchical multicomponent mixture prior for SNP effects that incorporates functional annotation data and allows for any distribution of SNP effects in each annotation.

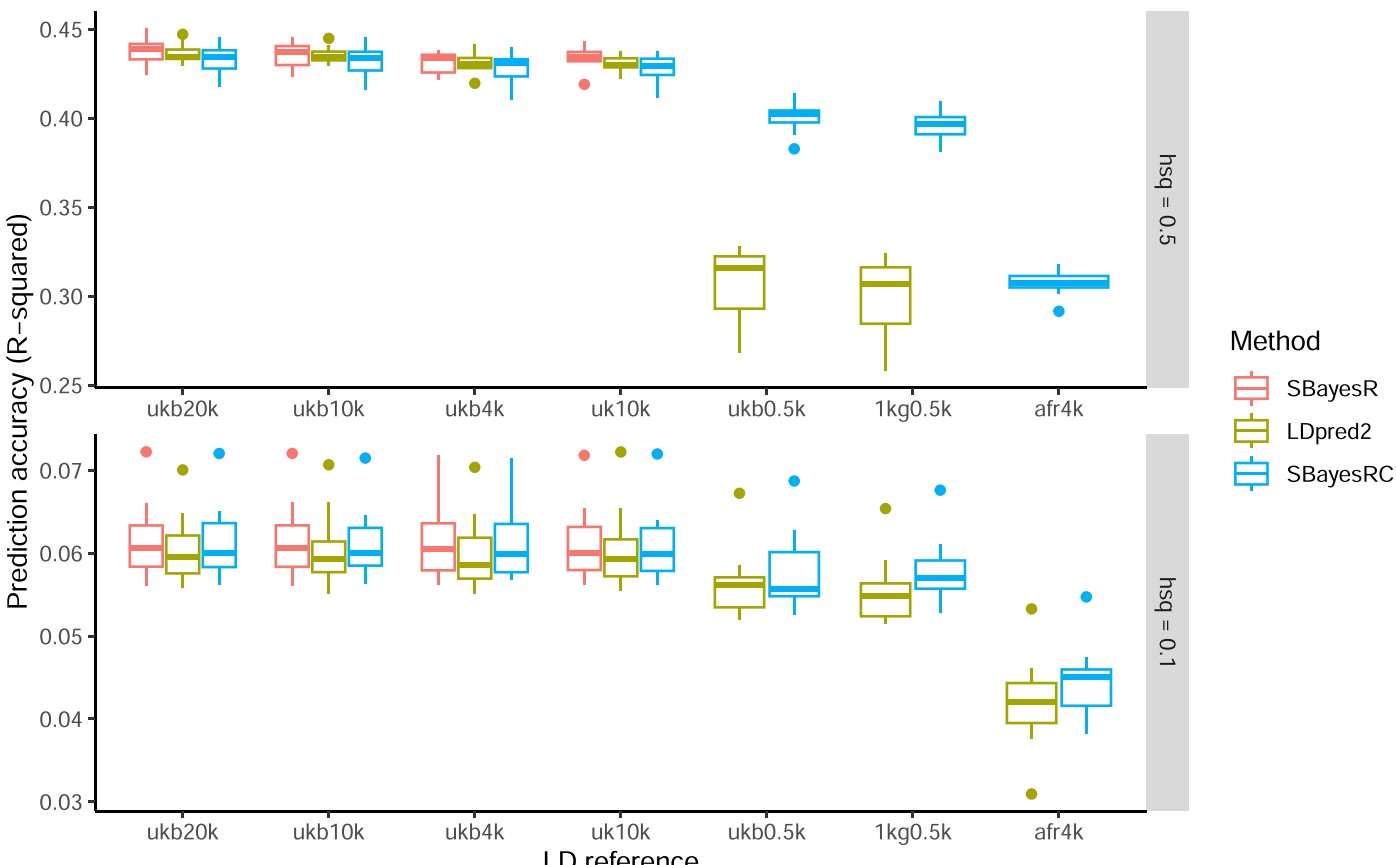

**Extended Data Fig. 2 | Robustness of SBayesRC to the choice of LD reference in simulation with HapMap3 SNP panel.** LD reference datasets included: ukb20k, 20,000 random sample from the UKB of European ancestry (EUR); ukb10k, 10,000 random sample from UKB EUR; ukb4k, 4,000 random sample from UKB EUR; uk10k, 3,642 unrelated samples from the UK10K dataset; ukb0.5k, 500 random sample from UKB EUR; 1kg0.5k, 494 unrelated samples from 1000GP EUR; afr4k, 4,000 random samples from the UKB of African ancestry (AFR). Each box plot shows the spread of data in 10 independent simulations; the line is the middle (median), the box covers the middle half (IQR), the whiskers extend to 1.5 times the IQR, and dots show outliers.

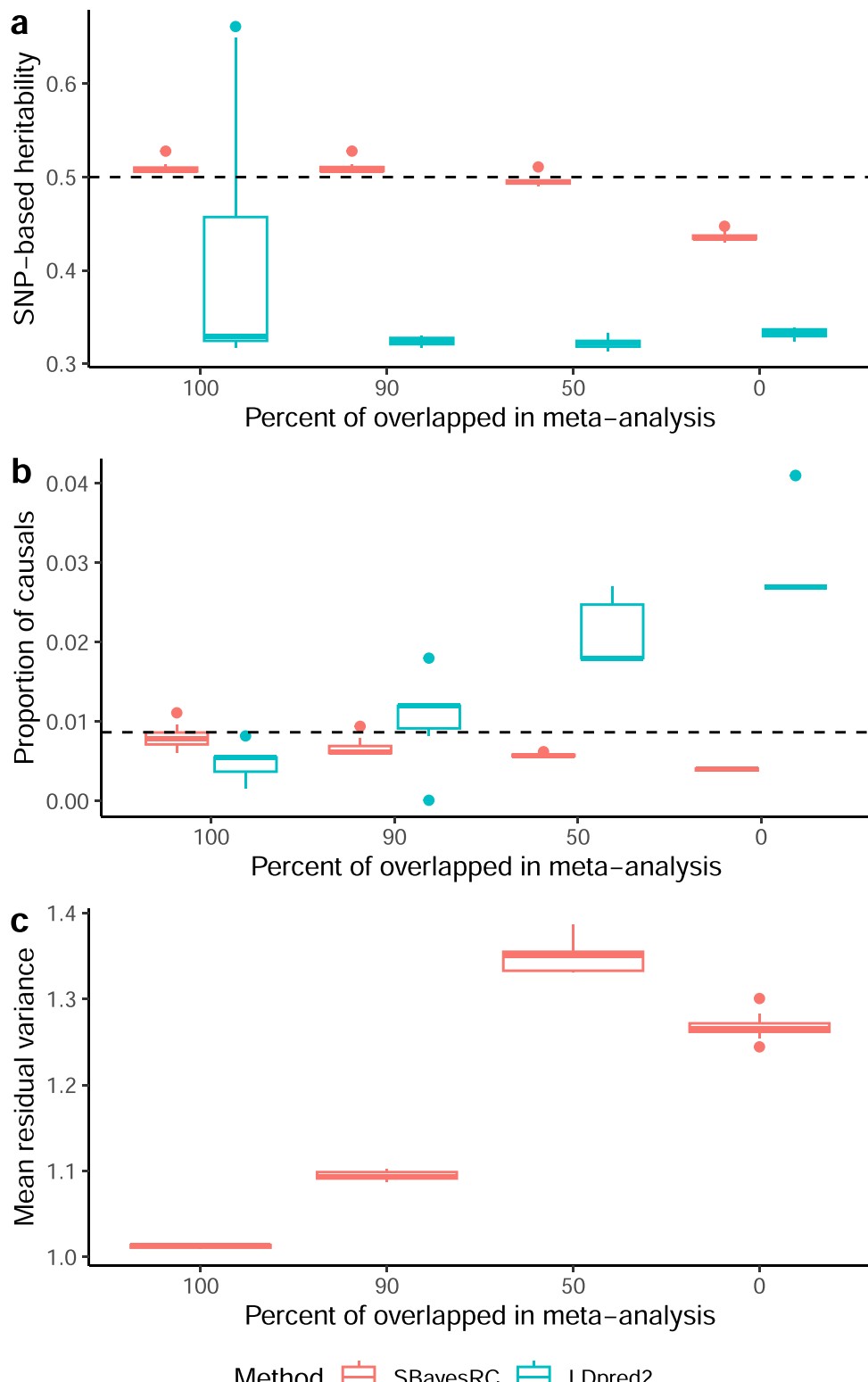

**Extended Data Fig. 3 | Parameter estimation from SBayesRC and LDpred2 using summary statistics from a meta-analysis of two simulated cohorts.** The proportion of overlapped SNPs between the two cohorts varied from 100 to 0. When the proportion of overlapping is less than 100, there existed unequal per-SNP sample sizes in the GWAS summary data. **a,b,** SBayesRC gave approximately unbiased estimates for SNP-based heritability (true value = 0.5) and polygenicity (true value = 0.01), whereas these estimates in LDpred2 were largely biased. **c,** The model misspecification affected the residual variance in SBayesRC, which is a nuisance parameter in the model. Each box plot shows the spread of data in 10 independent simulations; the line is the middle (median), the box covers the middle half (IQR), the whiskers extend to 1.5 times the IQR, and dots show outliers. The dashed line indicates the true value in simulation.

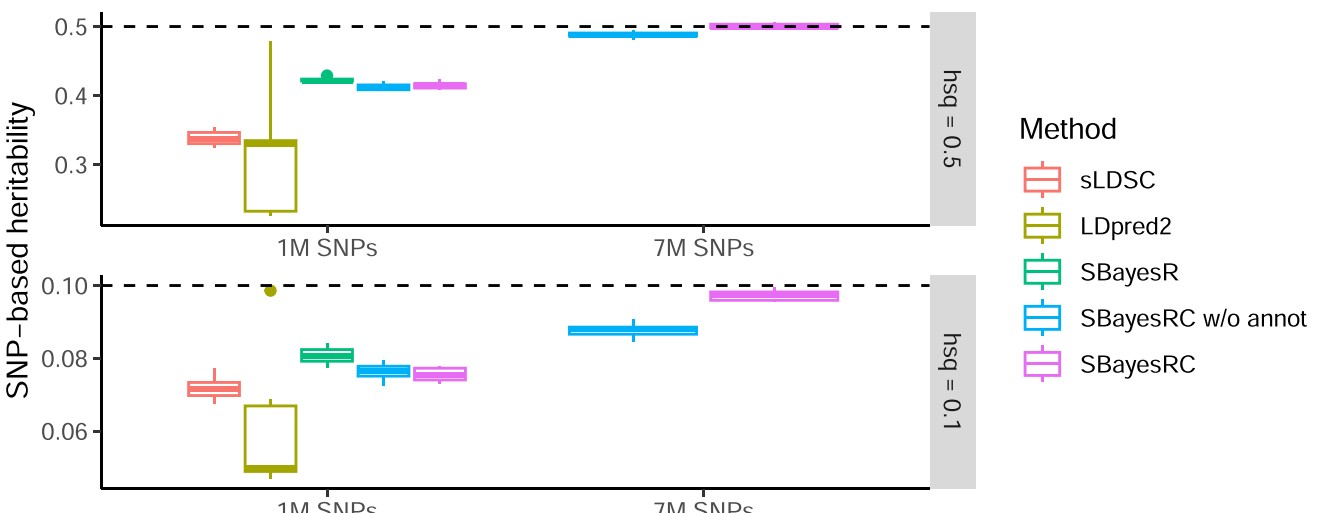

**Extended Data Fig. 4 | SNP-based heritability estimation from different methods and SNP panels for simulated traits.** Plots show a simulated trait with heritability of 0.1 (bottom) or 0.5 (top). The dashed line indicates the true value in the simulation. Each box plot shows the spread of data in 10 independent simulations; the line is the middle (median), the box covers the middle half (IQR), the whiskers extend to 1.5 times the IQR, and dots show outliers.

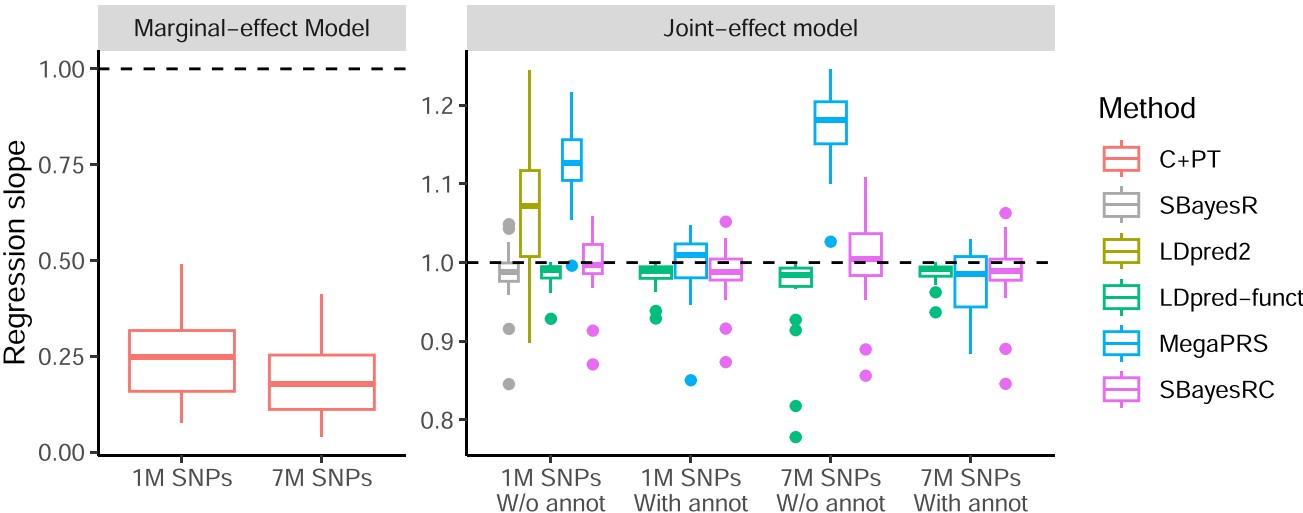

**Extended Data Fig. 5 | Mean regression slope from different methods for 28 independent traits across 10-fold cross-validations in the UKB unrelated sample of European ancestry.** Note that 8 traits in LDpred-funct and MegaPRS had a large regression slope (>2) and hence were removed from the LDpred-funct and MegaPRS column for better visibility of the other methods. The dashed line indicates the regression slope of 1. Each box plot shows the spread of data in 28 independent traits; the line is the middle (median), the box covers the middle half (IQR), the whiskers extend to 1.5 times the IQR, and dots show outliers. The values are shown in Supplementary Table 3.

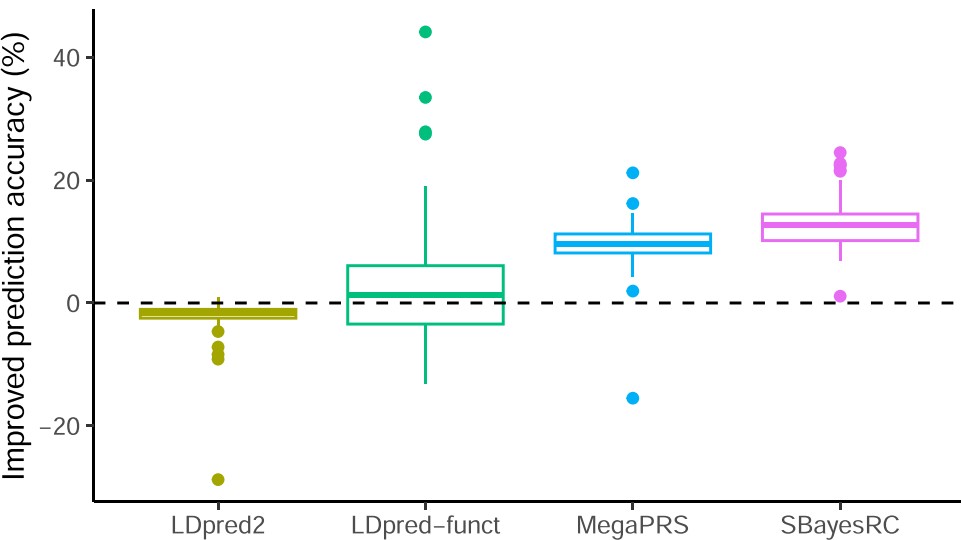

**Extended Data Fig. 6 | Relative prediction accuracy of different methods to SBayesR for 50 UKB traits.** Plots shows relative prediction accuracy of different methods to SBayesR using 1 M HapMap3 SNPs, averaged from 10-fold cross-validation for 50 UKB traits. Except for LDpred2 and SBayesR (HapMap3 SNPs without annotation), all other methods used imputed 7 M SNPs with functional annotations. The dashed line indicates the improved prediction accuracy equals to 0 (prediction accuracy equals to SBayesR using 1 M HapMap3 SNPs). Each box plot shows the spread of data in 50 traits; the line is the middle (median), the box covers the middle half (IQR), the whiskers extend to 1.5 times the IQR, and dots show outliers.

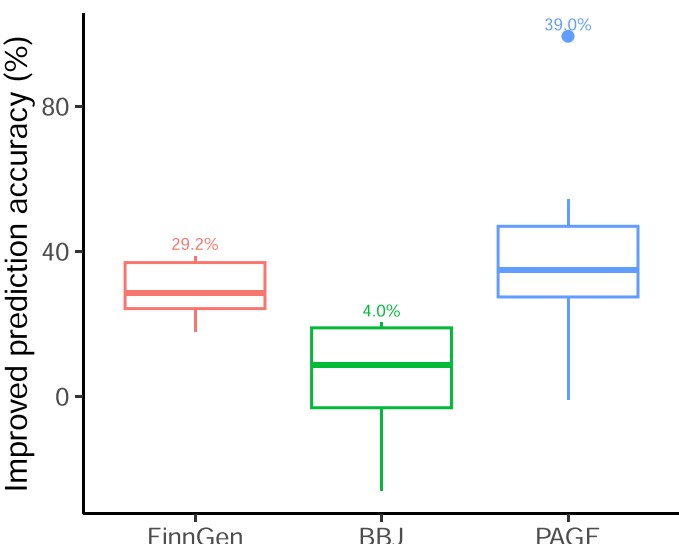

**Extended Data Fig. 7 | The improvement in prediction accuracy of SBayesRC relative to MegaPRS when an external LD reference independent of the GWAS sample was used in the analysis.** The improved prediction accuracy was calculated as $\left(\frac{R^2_{SBayesRC}-R^2_{MegaPRS}}{R^2_{MegaPRS}}\%\right)$. FinnGen ($n$ = 5 traits): GWAS summary data from FinnGen as training, a random sample of UKB UKB as LD reference, and the full UKB EUR as validation; BBJ ($n$ = 8 traits): GWAS summary data from BBJ as training, UKB EAS as LD reference, and UKB EAS as validation; PAGE ($n$ = 8 traits): GWAS summary data from PAGE (mixed AFR ancestry) as training, UKB AFR as LD reference, and UKB AFR as validation. Each box plot shows the spread of data; the line is the middle (median), the box covers the middle half (IQR), the whiskers extend to 1.5 times the IQR, and dots show outliers.

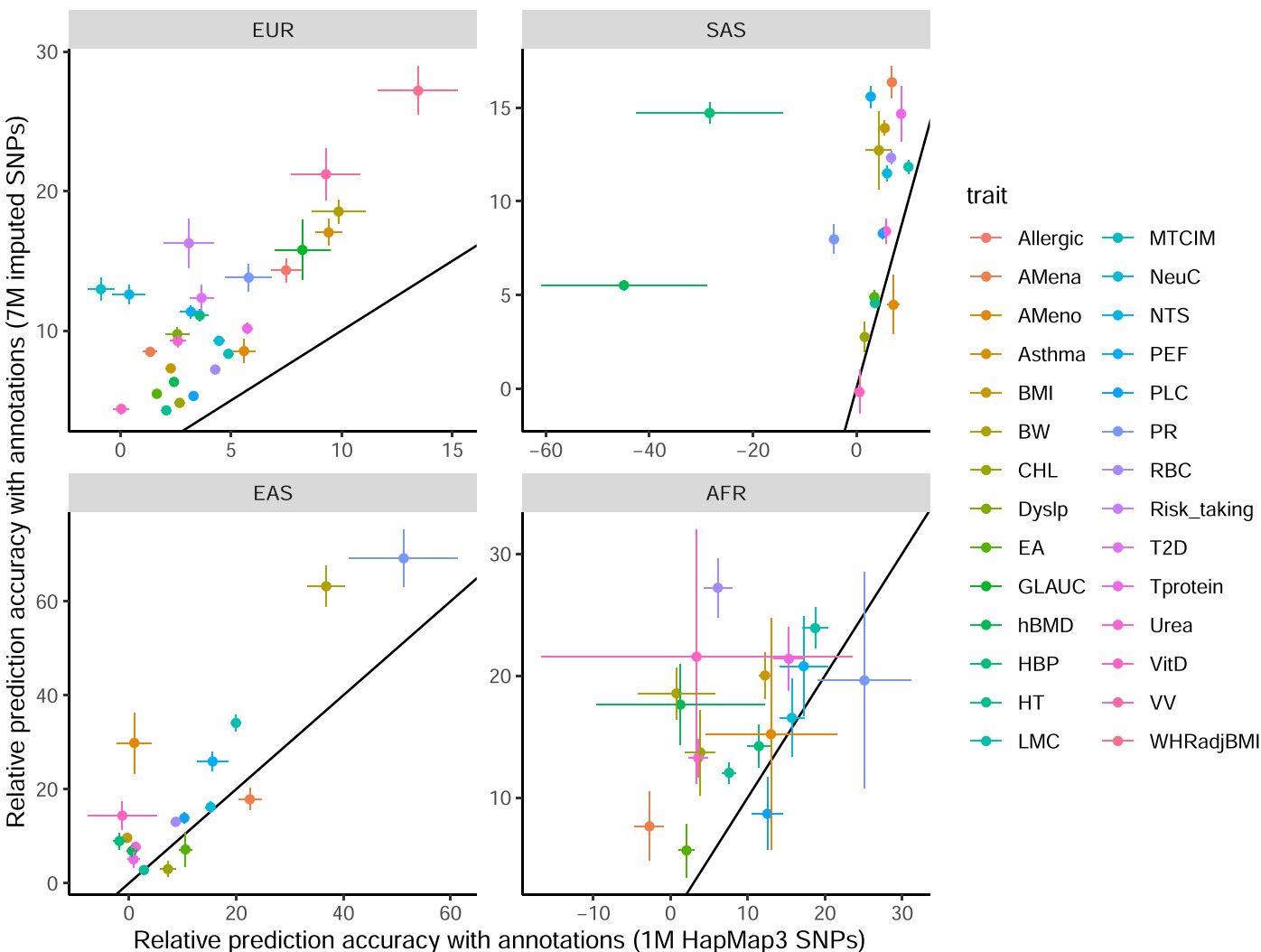

**Extended Data Fig. 8 | Interaction effects in all populations.** Comparison between 1 M HapMap3 SNPs and 7 M imputed SNPs for the improvement (%) in prediction accuracy for SBayesRC using annotations relative to SBayesRC without annotations. Results are from the 10-fold cross-validation in different ancestries in the UKB. Dots show the mean relative prediction accuracy and bars show the standard error estimated from the cross-validation. The black solid line indicates $y = x$.

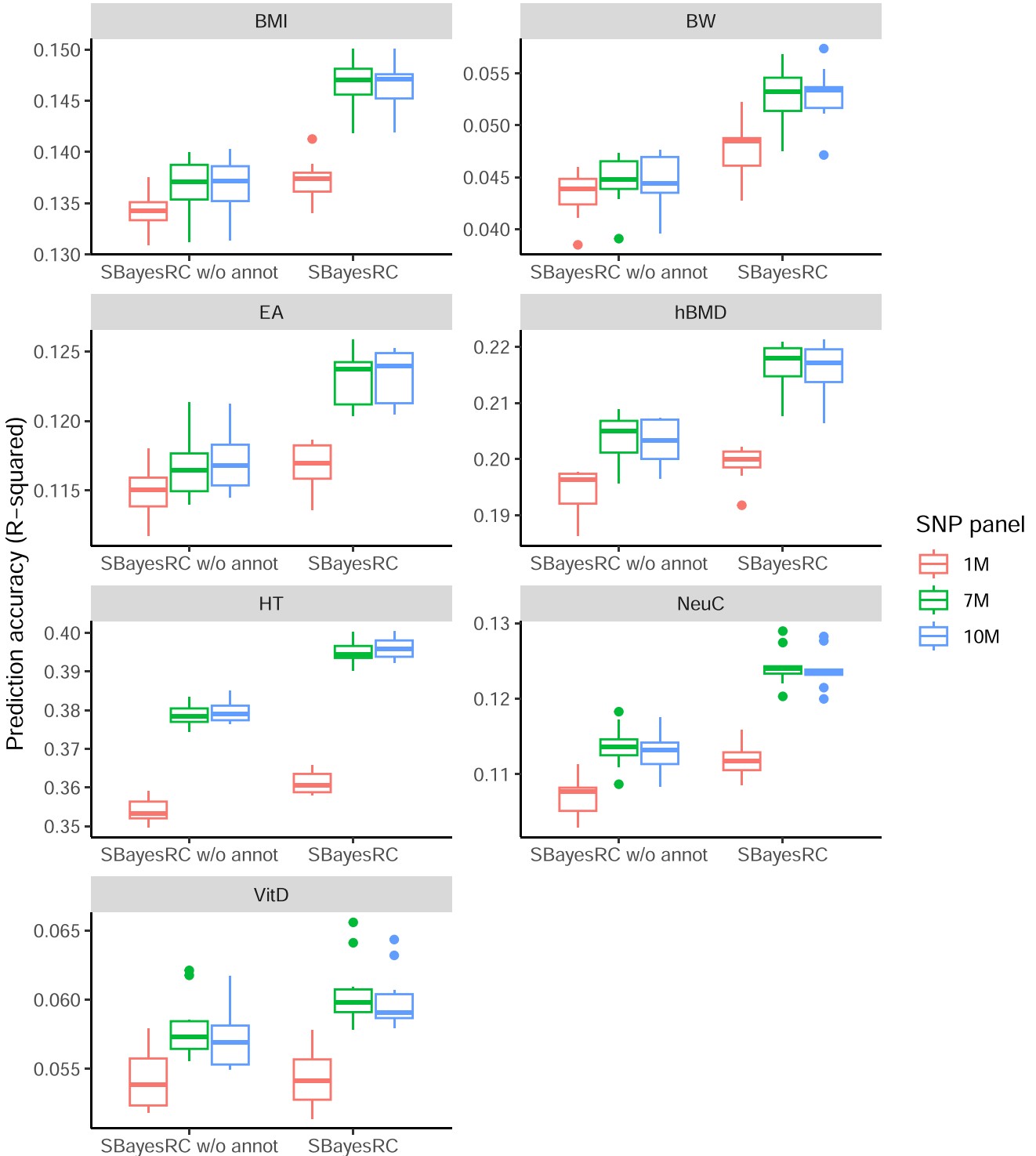

**Extended Data Fig. 9 | Prediction accuracy of SBayesRC.** Prediction accuracy of SBayesR (without annotation) or SBayesRC (with annotations) using 1 M, 7 M or 10 M common SNPs for 7 UKB traits. Each box shows the results of 10-fold cross-validation in the unrelated European sample. Trait acronym: BMI, body mass index; BW, birth weight; EA, educational attainment; hBMD, heel bone mineral density; HT, height; NeuC, neutrophile cell count; VitD, vitamin D level. Each box plot shows the spread of data in 10 cross-validation for that trait; the line is the middle (median), the box covers the middle half (IQR), the whiskers extend to 1.5 times the IQR, and dots show outliers.

**Extended Data Table 1 | Summary of prediction methods used in this study**

| Method | Features | Use annotations | Use all imputed SNPs | Require tuning data |
|---|---|---|---|---|
| C+PT[39] | Non-parametric method based on GWAS marginal effect estimates of a subset of independent SNPs obtained by LD clumping and p-value thresholding. | No | No | Yes |
| LDpred2[36] | Bayesian mixture model that fits SNPs jointly with a point-normal prior. More powerful than C+PT, accounting for sparse genetic architecture and LD between SNPs. LDpred2 has multiple models, including an infinitesimal model, grid of models, and automatic model, where the grid of models with tuning data is recommended. | No | No | Yes |
| LDpred-funct[28] | Stepwise method starting with an infinitesimal model with functionally informed priors and considering sparse genetic architecture empirically by regularizing SNP effects in bins of different magnitude using cross-validation. | Yes | Yes | Yes |
| MegaPRS[29] | Stepwise method. Step 1 estimates functional enrichment in SNP-based heritability using SumHer, an LD-weighted method for heritability estimation. Only SNPs with enriched per-SNP heritability are put forward to Step 2, where SNP joint effects are estimated from one of the models, e.g., LDAK-BayesR-SS, the recommended model similar to SBayesR. | Yes | Yes | No |
| SBayesR[34] | Multi-component mixture model assuming a finite mixture of normal distributions for SNP effects. Accounts for various genetic architectures. All parameters in the mixture distribution are estimated from data. | No | No | No |
| SBayesRC (this work) | Extension of SBayesR incorporating functional annotations and allowing for fitting all imputed SNPs jointly. Unified Bayesian framework to consider differences between annotations in both the proportion of causal variants and the magnitude of causal effect sizes. | Yes | Yes | No |
| PolyPred-S[15] | Stepwise method combining SNP effect estimates from a functionally informed fine-mapping approach for 18 million SNPs with those from SBayesR for 1 million HapMap3 SNPs. Flexible to the choice of the method generating the joint effect estimates of HapMap3 SNPs, with SBayesR recommended when LD reference sample closely matches the GWAS sample. Mainly for cross-population prediction, weighing the SNP effects from a sample of the target population. | Yes | Yes | Yes |
| PRS-CSx[14] | Multivariate method designed to improve cross-population prediction by jointly modelling summary statistics data from multiple ancestrally diverse populations. Borrows information across populations by a shared continuous shrinkage prior but still requires a tuning data to maximize the prediction accuracy in the target population. Does not consider functional annotations and is computationally challenging to fit more than 1 million HapMap3 SNPs. | No | No | Yes |
| SBayesRC-multi (this work) | Extension of SBayesRC to combine the SNP effect estimates from multiple populations after running SBayesRC in each population separately. Similar to PRS-CSx, a tuning sample of individual-level data is used to derive the final predictors for the target population. Unlike PRS-CSx, GWAS summary data from different populations are not jointly modelled. | Yes | Yes | Yes |

# Reporting Summary

## Statistics

For all statistical analyses, confirm that the following items are present in the figure legend, table legend, main text, or Methods section.

| n/a | Confirmed | |
|---|---|---|
| ☐ | ☒ | The exact sample size (*n*) for each experimental group/condition, given as a discrete number and unit of measurement |
| ☒ | ☐ | A statement on whether measurements were taken from distinct samples or whether the same sample was measured repeatedly |
| ☐ | ☒ | The statistical test(s) used AND whether they are one- or two-sided *Only common tests should be described solely by name; describe more complex techniques in the Methods section.* |
| ☐ | ☒ | A description of all covariates tested |
| ☐ | ☒ | A description of any assumptions or corrections, such as tests of normality and adjustment for multiple comparisons |
| ☐ | ☒ | A full description of the statistical parameters including central tendency (e.g. means) or other basic estimates (e.g. regression coefficient) AND variation (e.g. standard deviation) or associated estimates of uncertainty (e.g. confidence intervals) |
| ☐ | ☒ | For null hypothesis testing, the test statistic (e.g. *F*, *t*, *r*) with confidence intervals, effect sizes, degrees of freedom and *P* value noted *Give P values as exact values whenever suitable.* |
| ☐ | ☒ | For Bayesian analysis, information on the choice of priors and Markov chain Monte Carlo settings |
| ☒ | ☐ | For hierarchical and complex designs, identification of the appropriate level for tests and full reporting of outcomes |
| ☐ | ☒ | Estimates of effect sizes (e.g. Cohen's *d*, Pearson's *r*), indicating how they were calculated |

*Our web collection on statistics for biologists contains articles on many of the points above.*

## Software and code

Policy information about availability of computer code

| Data collection | No data collection software was used in this study. The data was described in the Methods of the manuscript, which was collected by multiple Biobanks. |
|---|---|
| Data analysis | We used GCTA v1.93.2, PLINK2 v2.00a2.3 and R 4.0.4 for the quality control of individual level data and generate the summary statistics as described in the Methods section. We obtained the SNP weights from summary statistics via SBayesRC-v0.2.0(https://github.com/zhilizheng/SBayesRC), PLINK v1.90b6.21, LDpred2 v1.8.1, GCTB v2.03, PolyPred (Feb 1, 2022), LDpred-funct (Aug 30, 2021), MegaPRS (v5.2) and PRS-CSx (Jun 29, 2022). The polygenic scores were calculated by PLINK2 v2.00a2.3 from the weights. Refer to the links in the main text to all the analysis tools and Methods section for details. |

For manuscripts utilizing custom algorithms or software that are central to the research but not yet described in published literature, software must be made available to editors and reviewers. We strongly encourage code deposition in a community repository (e.g. GitHub). See the Nature Portfolio guidelines for submitting code & software for further information.

## Data

Policy information about availability of data

All manuscripts must include a data availability statement. This statement should provide the following information, where applicable:

- Accession codes, unique identifiers, or web links for publicly available datasets
- A description of any restrictions on data availability
- For clinical datasets or third party data, please ensure that the statement adheres to our policy

> We have updated the Data Availability statement in the manuscript:
> The UK Biobank data, UK10K and LifeLines are available through formal application to the UK Biobank (http://www.ukbiobank.ac.uk), UK10K (https://www.uk10k.org/data_access.html) and LifeLines (https://www.lifelines.nl/researcher/how-to-apply). The summary data and PGS weights from SBayesRC for the 28 approximately independent UKB traits can be found at https://cnsgenomics.com/software/gctb/#Download. All the other datasets used in this study are available in the public domain. 1000 Genomes: https://www.internationalgenome.org/data/; FinnGen: https://www.finngen.fi/en/access_results (version: R8); BBJ: https://biobankjp.org/en/ ; PAGE: https://www.ebi.ac.uk/gwas/publications/31217584

## Human research participants

Policy information about studies involving human research participants and Sex and Gender in Research.

| | |
|---|---|
| Reporting on sex and gender | We obtained self-reported sex from the UK biobank and LifeLines cohort and confirmed this information by genotype data. We performed rank based inverse transformation on quantitative trait phenotypes within each sex to adjust for the difference in mean and variance between sex. Hence, our findings apply to both sexes. No individual level data including sex need to be reported in this study. The consent of using the data was approved by the data provider. |
| Population characteristics | UK Biobank is a large population-based cohort with nearly 500,000 participants, whose age ranged between 40 and 69 at recruitment, from 4 ancestries including European, East Asia, African, and South Asia. (see https://www.ukbiobank.ac.uk/key-documents/ for more details). Lifelines is a large, multi generational cohort study from northern population of Netherlands. The cohort included participants from three generations. The participants' age ranged between 0 and 93 in this cohort. The Biobank Japan (BBJ) is a cohort of patients diagnosed with any of the 47 target diseases who were enrolled in Japan. The average age was 62.7 years for men and 61.5 years for women at recruitment. The FinnGen project is a continuous research initiative that utilizes samples sourced from a nationwide network of Finnish biobanks and digital healthcare data retrieved from national health registers. Isolated Finn populations that have undergone recent bottlenecks can harbor harmful alleles predisposing to disease at much higher frequencies than what is expected in larger and older outbred populations, due to heightened genetic drift. The participants age ranged between 0 and 108 in this cohort. The Population Architecture using Genomics and Epidemiology (PAGE) study performed GWAS of 26 phenotypes in 49,839 non-European individuals. PAGE examined putative causal genetic variants across multi-ethnic and admixed populations: African Americans, Asian Americans, American Indians, European Americans, Hispanic Americans, and Native Hawaiians from four groups representing nine large U.S.-based cohorts. The distribution of covariates is unknown in some cohorts from the data available publicly (e.g. HCHS/SOL: aged 18-74 years; WHI: all women, 50-79 years of age; MEC: 45-75 years of 85 age at baseline). |
| Recruitment | Recruitment of the samples has been described in previous studies. We do not produce new data and have cited the previous work. |
| Ethics oversight | University of Queensland Human Research Ethics Committee B |

Note that full information on the approval of the study protocol must also be provided in the manuscript.

# Field-specific reporting

Please select the one below that is the best fit for your research. If you are not sure, read the appropriate sections before making your selection.

☒ Life sciences ☐ Behavioural & social sciences ☐ Ecological, evolutionary & environmental sciences

For a reference copy of the document with all sections, see nature.com/documents/nr-reporting-summary-flat.pdf

# Life sciences study design

All studies must disclose on these points even when the disclosure is negative.

| | |
|---|---|
| Sample size | The sample size for training was determined by the maximum number of samples from European ancestry with imputed genotype data in the UK Biobank and the summary-level data available from GIANT and Biobank Japan. After the quality control, the UK Biobank contains unrelated individuals of 4 ancestries, i.e., European ( N= 347,800), East Asian ( N=2,252), South Asian ( N=9,436) and African ( N=7,006) and the sample size for selected traits ranged from 110,334 to 360,503. The LifeLines cohort contains 11,842 samples of European ancestry after quality |

control. Summary-level data from the GIANT consortium (Yengo et al 2018) has the median per-SNP sample size of 704,810 for height and 688,632 for BMI. Summary data for diastolic blood pressure (Evangelou et al) had a sample size of 756,595, and type 2 diabetes (Xue et al) has a sample size of 62,892. The sample size in selected traits in Biobank Japan ranged from 82,810 to 165,056; the sample size for PAGE ranged from 28,534 to 49,781; sample size for FinnGen range from 163,394 to 342,439. We used those dataset as training or validation, which is the largest sample size current available. We also investigated the improvement in prediction accuracy of our method along with different sample sizes by down sampling in the UK Biobank, the prediction accuracy will increase along with more samples. Current data is sufficient to demonstrate the improved prediction accuracy.

**Data exclusions**

UK Biobank:
We removed SNPs with MAF < 0.01, Hardy-Weinberg Equilibrium test P < 10e-10, imputation info score < 0.6 in European samples. We used the GCTA software to remove the cryptic relatedness in the UKB based on the HapMap3 SNPs in each population. The samples were pruned by estimated relatedness larger than 0.05, keeping the unrelated samples only. We also removed the samples with mismatched sex information in phenotype and genotype, and samples withdrawn from the participation.
We removed the SNPs that are not in common among UKB, the annotation baseline model BaselineLD v2.2 and LifeLines cohort, resulting in 7,356,518 imputed common SNPs and 1,154,522 HapMap3 common SNPs. The 55 traits with relatively large sample size (N>110,000) were extracted from all 4 ancestries which were further pruned to 28 traits by absolute phenotypic correlation of 0.1. The traits with a mean prediction accuracy < 0.01 among methods were removed. The phenotypes with continuous values were filtered within the range of mean +/- 7SD.
LifeLines:
We removed SNPs with imputation info score < 0.3, MAF < 0.0001 and HWE < 1e-6. We removed the sample with age < 20 years old and removed samples with the phenotypic value (height and BMI) out of the range of mean +/- 5SD. We further removed the related samples and retained 11,842 unrelated samples.
Public data from GWAS meta-analysis for height, BMI, diastolic blood pressure, and type 2 diabetes:
We removed the SNPs with per-SNP sample size out the range of mean +/- 3 SD and the difference in allele frequency between GWAS and LD reference samples larger than 0.2. The summary data was further QCed by DENTIST to filter the SNPs with potential errors, removed the SNPs with P_DENTIST < 5e-8 and P_GWAS > 0.01.
BioBank Japan:
We matched the SNPs in the summary data from BBJ and UKB and removed SNPs with MAF < 0.005 in either population. After QC, 4,906,538 SNPs remained with functional annotations, of which 1,011,961 SNPs were in the HapMap3 panel.
PAGE:
We matched the SNPs in the summary data from PAGE and UKB and removed SNPs with MAF < 0.005 in either population. After QC, 6,064,174 SNPs remained, of which 1,131,955 SNPs were in the HapMap3 pane

**Replication**

We replicated the simulation 10 times in each scenario to evaluate the performance of our method. For real data analysis, we performed 10 cross validation to obtain the average prediction accuracy across validation folds. For cross biobank and cross ancestry prediction, no replication was performed because we used the whole Biobank or summary-level data to maximize the statistical power.

**Randomization**

Randomization for sample collection was not relevant to this study because we performed the analysis in publicly available data. We randomized in the simulation and cross validation in real data, with details described in the Methods section.

**Blinding**

Blinding was not relevant to this study, because we performed the analysis in publicly available data, We did not use any study design that required blinding, with the details described in the Methods section.

# Reporting for specific materials, systems and methods

We require information from authors about some types of materials, experimental systems and methods used in many studies. Here, indicate whether each material, system or method listed is relevant to your study. If you are not sure if a list item applies to your research, read the appropriate section before selecting a response.

## Materials & experimental systems

| n/a | Involved in the study |
|---|---|
| ☒ | Antibodies |
| ☒ | Eukaryotic cell lines |
| ☒ | Palaeontology and archaeology |
| ☒ | Animals and other organisms |
| ☒ | Clinical data |
| ☒ | Dual use research of concern |

## Methods

| n/a | Involved in the study |
|---|---|
| ☒ | ChIP-seq |
| ☒ | Flow cytometry |
| ☒ | MRI-based neuroimaging |

