## [Peer Review File · Nature Genetics]

Peer Review Information

Manuscript Title: Leveraging functional genomic annotations and genome coverage to improve polygenic prediction of complex traits within and between ancestries

Corresponding author name(s): Dr Jian Zeng ,Dr Zhili Zheng

Reviewer Comments & Decisions:

Decision Letter, initial version:

1st December 2022

Dear Jian,

Your Analysis "Leveraging functional genomic annotations and genome coverage to improve polygenic prediction of complex traits within and between ancestries" has been seen by three referees. You will see from their comments below that, while they find your work of interest, they have raised several relevant points. We are interested in the possibility of publishing your study in Nature Genetics, but we would like to consider your response to these points in the form of a revised manuscript before we make a final decision on publication.

To guide the scope of the revisions, the editors discuss the referee reports in detail within the team, including with the chief editor, with a view to identifying key priorities that should be addressed in revision, and sometimes overruling referee requests that are deemed beyond the scope of the current study. In this case, we ask that you address all technical queries with suitable clarifications and additional analyses where required and that you extend the benchmarking analyses to include MegaPRS and a broader set of traits. We hope you will find this prioritized set of referee points to be useful when revising your study. Please do not hesitate to get in touch if you would like to discuss these issues further.

We therefore invite you to revise your manuscript taking into account all reviewer and editor comments. Please highlight all changes in the manuscript text file. At this stage we will need you to upload a copy of the manuscript in MS Word .docx or similar editable format.

*2) If you have not done so already please begin to revise your manuscript so that it conforms to our Analysis format instructions, available [here](http://www.nature.com/ng/authors/article_types/index.html). Refer also to any guidelines provided in this letter.

[redacted]

We hope to receive your revised manuscript within 8-12 weeks. If you cannot send it within this time, please let us know.

Sincerely,
Kyle

Kyle Vogan, PhD
Senior Editor
Nature Genetics
<https://orcid.org/0000-0001-9565-9665>

Referee expertise:

Referee #1: Genetics, statistical methods, polygenic prediction

Referee #2: Genetics, statistical methods, polygenic prediction

Referee #3: Genetics, statistical methods, polygenic prediction

Reviewers' Comments:

Reviewer #1:
Remarks to the Author:

In this paper, Zheng et al introduced SBayesRC, an extension to SBayesR. Overall, this is a very well written paper, with clear and detail explanation of the analyses and the methodology. The improved computational efficiency, and the ability to perform cross-ancestry PRS analyses without necessitating a tuning sample, is welcoming. I have only some minor comments:

1. One of the main claims of the paper is that there is a significant interaction between SNP density and annotation information on the predictive performance. However, I am unsure if the authors performed a formal statistical test to establish the interactive effect of SNP density and whether annotations were included or not. The p-value cited seems to be the improvement of using 7M imputed SNPs vs 1M HapMap 3 SNPs instead of the interactive effect of SNP density and annotation.
2. It is quite interesting to see that LDpred-func performs worse than LDpred2 in most if not all the analyses presented in the current paper. While I understand that LDpred-func is an extension of LDpred instead of LDpred2, which might partially explain the difference in performance, I still find it surprising how bad it performs despite incorporating the annotation information.
3. For the trans-ancestry analyses, which LD matrix do SBayesR and SBayesRC use for the PRS calculation? Do you match the LD population to the GWAS samples (EUR) or the target samples?
4. In Figure 4, when incorporating more SNPs into the model in EAS samples, the performance of SBayesRC without annotation seems to be worse than SBayesR. Is there any possible explanation to this observation?
5. While trans-ancestry analyses aren't the focus of the current paper, it is still of interest to see the performance of SBayesRC vs PRS-CSx in African samples, which is known to have the worst portability in PRS analyses. Summary statistics from PAGE might be of use, though this might be something for a future paper.

Some minor recommendation:

1. Considering the bottleneck of the current algorithm is due to the eigenvalue decomposition algorithm, a possible optimization for future version of SBayesRC might be the use of the Spectra C++ library, which allows for a quick calculation of eigen value and eigen vector on large sparse matrix.
2. While I can find the R-version of the SBayesRC algorithm on the github page, it does not seem clear to me if the algorithm is implemented in the GCTB software. In the documentation on the GCTB website, while there is an option `--annot`, which I assume will run SBayesRC instead of SBayesR, the documentation stated that "The current version only allows binary annotations..." which seems to contradict to what was stated in the current paper, which stated that SBayesRC can handle continuous annotation. Maybe the authors should update the documentations of GCTB to reflect this change?
3. A minor nitpicks on the coloring scheme: it would be nice if the coloring of each tools are consistent across different figures (e.g. LDpred2, SBayesR, SBayesRC, PolyPred-S).

Reviewer #2:

Remarks to the Author:

The authors present an extension of the well established SBayesR methodology for PRS construction. This method adds functional labels to refine the SNP weights, which should in principle (and as shown here, in practice) improve fine-mapping, and therefore PRS robustness across ancestry groups.

As far as I can tell, SBayesR is used relatively widely and several other PRS prediction tools have developed their own version of functionally informed fine-mapping. It therefore makes sense to see SBayesR receive a similar upgrade. While the motivation and purpose and obviously well described in the literature, there are many useful insights from this paper, in particular the contribution of SNP density and its interaction with functional priors, together with added cross ancestry robustness. The analysis is thorough, exhaustive and rigorous. I cannot see major flaws in the arguments. The open source availability of the code is noted and much appreciated.

Reading the paper, my major interrogation lies in my limited understanding of the intuition behind the ability of the model to learn functional weights from the data. The principle of SBayesR, as far as I understand it, consists of reducing the complex SNP diversity using a low rank transformation that, in effect, collapses groups of variants in high LD into independent variables in the model. The idea behind SNP functional data is of course the fact that even within an LD block, different variants can have very different functional annotations, and that should drive the choice of the variant receiving the higher weight. So I find it surprising that this low rank reduction can still enable the Bayesian machinery to learn about the different weights associated with functional annotations. Obviously it does, as evidenced in Figure 7c, but I would like the authors to help me understand why this training can work.

A second comment - and I appreciate it may just be me - is that I am struggling to interpret performance measured as r^2 for binary traits. These are typically measured using AUC, Harrel's C or (better in my opinion) effect size per SD of the PRS (ideally conditional on age, sex and PCs). If I could encourage the authors to adapt ST3 to show these metrics for binary traits, that would be much

appreciated.

Related to this point of metrics for performance evaluation, the r^2 value associated with the PRS for height is massively affected by the inclusion of sex as a covariate in the model (because sex explains about 50% of the height variance, adding it as a covariate doubles the proportion of the residual variance explained by the PRS). I think this is what is stated at line 736 in the Methods section, but I would make it clear in the main text because, as far as height is concerned, this generates substantial confusion in the literature. I would double check that the statements on line 277 comparing the PRS variance to the theoretical variance explained by SNPs are not comparing oranges and apples (i.e. one includes sex as a covariate and one does not). The large delta in these metrics between BMI and height makes me suspicious.

I would also tone down comments on the convergence issues observed in LDpred2 (line 180). These PRS tools typically require some minimum tuning and optimisation. Clearly, as authors of SBayes-RC, the authors will know how to make the most of the tool. My guess is that the LDpred2 authors (and I am not one of them!) would find a way to reach convergence. Hence, this comparison statement may come across as slightly unfair - though I appreciate that the authors also consider SBayesR, with extensive SBayesRC authorship overlap, so there is no doubt an argument for added robustness here.

I also would like to see the authors discuss the need for cross-validation in the UKB work. This is quite a substantial additional computational work, and the UKB is large enough to set aside a large training set while retaining excellent training sample size, especially for quantitative traits. Could I ask the authors to motivate that choice? Is that simply to extract as much data as possible from UKB or is there a rationale around added stability through that 10 fold procedure? Especially as the authors only use $\sim 342K$ unrelated individuals, there are substantial numbers of EUR ancestry individuals that remain usable, and a minimum relatedness cutoff would yield a very large testing set with less than measurable bias associated with relatedness to training in my experience.

I have a minor interpretation issue with the statement starting at line 361, in the results section. The authors make the case that annotation benefits will be maximised when working with sequence data. This seems contradictory with the fact that the authors see an asymptote in performance when a larger set of $\sim 10M$ SNPs are used. The contradiction is resolved by arguing that this may be a limit of the imputation accuracy. That may be true but one could also argue in the other direction and say that the asymptote is reached because most causal variants are contained in the 7M set. In any case this is a discussion statement, not a result and I would be cautious with this as the added value of sequence data for PRS construction remains unclear.

Just a minor technicality but my pdf version of Supp Tables only includes ST5. That confused me for a while until I found the remaining table in the excel doc. Something to check...

Vincent Plagnol

Reviewer #3:
Remarks to the Author:

This manuscript proposes a method SBayesRC to compute polygenic risk scores (PRS). SBayesRC

extends SBayesR to use more SNPs and functional annotations. Simulation and real data analysis revealed that SBayesRC improves the prediction accuracy by leveraging the interaction between SNP density and functional annotation. Overall, I think SBayesRC is a solid method and the idea of using the low-rank model is clever. My main concern is about (1) the choice on how to model functional annotation with effect sizes and (2) lack of comparison with MegaPRS to show the improvement. Below are my comments.

Major:

1. SBayesRC assumes that the probability of distribution membership π_{jk} is related to the functional annotation. Another way to model the functional annotation is to assume that the variance of effect sizes (heritability) for each SNP is additively related to the functional annotation (e.g. S-LDSC <https://www.ncbi.nlm.nih.gov/pmc/articles/PMC4626285/> and megaPRS <https://www.nature.com/articles/s41467-021-24485-y#Sec8>). The authors could discuss why they prefer to use the first approach to model the functional annotation.
2. At the same point, for the simulation scenario incorporating annotation data, the SNP probability of membership was linked with the annotation. It would be helpful to investigate the performance of SBayesRC when the variance of effect sizes is additively linked to the functional annotation.
3. SBayesRC assumes that the scaling factor of five distributions as the mixture components is fixed as $\gamma = [0, 0.001, 0.01, 0.1, 1]$. In simulation, the authors specified the correct γ . I was wondering whether SBayesRC would be robust to the misspecification of γ .
4. To my knowledge, SBayesR is not the first method that utilizes the functional annotation and use more SNPs to compute PRS. MegaPRS also uses functional annotation, $\sim 7.5M$ imputed SNPs (<https://www.nature.com/articles/s41467-021-24485-y>) and has similar assumptions on effect sizes (multi-component mixture priors). It would be helpful if the authors could compare SBayesRC with MegaPRS in simulations and real data analysis.
5. In my opinion, out-of-sample prediction performance based on external summary statistics is the most important and commonly used evaluation approach for the benchmark of PRS methods. Based on my experience, SBayesR can sometimes have convergence problems when applied to external summary statistics. Currently only 2 traits (height and BMI) were analyzed in the benchmark. It would be more convincing if the authors could demonstrate that SBayesRC outperforms other methods on more traits using external summary statistics.
6. Have the authors considered using AFR instead of EAS summary statistics (e.g. PAGE) to derive PRS for trans-ancestry prediction in AFR populations?

Minor:

1. In Figure 1a, why is Anno3 of "causal variant unobserved" different with Anno3 of "causal variant observed"?
2. Results from PRS-CSx should be included in Figure 4.
3. SBayesRC uses 5 mixture components. I was wondering whether increasing or decreasing the

number of components could affect the performance.

4. Tuning parameters of LDpred2 and PRS-CSx should be listed in the method section.

5. In Figure 7c, what does the x-axis of "zero effect" mean? Why do some categories have negative values?

Author Rebuttal to Initial comments

We would like to express our gratitude for the constructive comments from the three reviewers. In this revision, we have now applied SBayesRC to 50 complex traits covering a diverse range of trait categories and genetic architectures, providing a comprehensive representation of human complex traits. We have extended our analysis to include MegaPRS for method comparison through additional simulation and real data analyses. We have also included more traits for out-of-sample prediction using external GWAS summary statistics. We have improved the robustness of our method by implementing a pseudo-tuning approach, leading to an enhanced performance. Moreover, we evaluated our method for the analysis of samples of African ancestry using PAGE data. Please find below our point-to-point response to the reviewer's comments in blue.

Reviewer #1:

Remarks to the Author:

In this paper, Zheng et al introduced SBayesRC, an extension to SBayesR. Overall, this is a very well written paper, with clear and detail explanation of the analyses and the methodology. The improved computational efficiency, and the ability to perform cross-ancestry PRS analyses without necessitating a tuning sample, is welcoming. I have only some minor comments:

1. One of the main claims of the paper is that there is a significant interaction between SNP density and annotation information on the predictive performance. However, I am unsure if the authors performed a formal statistical test to establish the interactive effect of SNP density and whether annotations were included or not. The p-value cited seems to be the improvement of using 7M imputed SNPs vs 1M HapMap 3 SNPs instead of the interactive effect of SNP density and annotation.

We thank the referee for their positive and constructive comments on our paper. In the original manuscript, we first quantified the improvement in prediction accuracy due to the use of annotation information by calculating the relative prediction accuracy from the full model that includes annotations to the basic model that excludes annotations ($\delta^2 = \frac{R_{Full}^2 - R_{Basic}^2}{R_{Basic}^2}$) at each SNP density level. Then, we evaluated the interaction between SNP density and annotation information by comparing δ^2 between 7M imputed SNPs (δ_{7M}^2) and 1M HapMap3 SNPs (δ_{1M}^2). If the benefit of including annotations is

independent of SNP density (i.e., no interactive effect between SNP density and annotation information), δ_{7M}^2 is expected to be equal to δ_{1M}^2 (i.e., equal amount of improvement in prediction accuracy no matter 7M or 1M SNPs are used). The original Figure 5 showed the average values of δ_{7M}^2 and δ_{1M}^2 across traits for different data sets, including simulation data sets with different heritability settings and the UKB data sets with different ancestries. We have now improved the clarity by showing the results of δ_{7M}^2 and δ_{1M}^2 in the European ancestry for each of the UKB traits analysed in the revised Figure 5, and those from the other ancestries in new Supplementary Figure 19. It can now be clearly seen from the revised Figure 5 that δ_{7M}^2 was significantly greater than δ_{1M}^2 in all 28 independent traits, mostly with a 2-fold difference or more, indicating a strong interaction between SNP density and annotation information. We further formally tested this interaction effect by fitting the indicator variables for SNP density and annotation data as well as their product to the prediction accuracy for each trait. To account for the variability in prediction accuracy between traits because of trait heritability, the prediction R^2 from different scenarios (regarding SNP density level and the use of annotations) for each trait was scaled relative to that using HapMap3 SNPs without annotations. Our test result showed that the interaction effect was highly significant (P value = 6.7×10^{-7}), in addition to the significant main effects (P value for SNP density = 4.2×10^{-4} and P value for annotation information = 1.1×10^{-5}). The interaction effect was also significant in MegaPRS ($P_{\text{Interaction}} = 1.1 \times 10^{-5}$) and LDpred-funct ($P_{\text{Interaction}} = 0.048$) (Revised Figure 3a), suggesting that this is capturing a biological signal independent of the prediction methods. We have included the results in the main text (lines 375-392 in Results, lines 498-530 in Discussion and lines 882-897 in Methods).

2. It is quite interesting to see that LDpred-func performs worse than LDpred2 in most if not all the analyses presented in the current paper. While I understand that LDpred-func is an extension of LDpred instead of LDpred2, which might partially explain the difference in performance, I still find it surprising how bad it performs despite incorporating the annotation information.

We thank the referee for raising this point and have examined the LDpred-funct analysis in more details. In the LDpred-funct paper (Marquez-Luna et al 2021), the authors compared LDpred-funct with LDpred and SBayesR but did not include LDpred2 for comparison. They reported, in the abstract, that LDpred-funct using 6M SNPs with annotations attained a 4.6% relative improvement in average prediction accuracy compared to SBayesR using 1M SNPs. This is generally consistent with the result in our analysis that LDpred-funct using 7M SNPs with annotations on average gave 3.5% higher relative prediction accuracy than the standard SBayesR with 1M SNPs (in our SBayesR analysis the 3cM banded matrices was used, same as in LDpred2, which is slightly superior to the shrunk matrices used in the analysis of Marquez-Luna et al). Our investigations show that the superiority of LDpred-funct arises from the combination of using high-density SNPs and annotation data. However, the way LDpred-funct models the data is suboptimal. This can be seen from the result that, when using 1M SNPs with annotations, LDpred-funct is worse than SBayesRC and that, when using 1M SNPs without annotations, LDpred-funct is worse than either SBayesR or LDpred2 on average. Our explanation is that unlike LDpred2 and SBayesR where SNP effects are assumed to have a mixture distribution, which has been shown to be a better model for the genetic architecture for complex traits in the literature (Moser et al 2015; Lolyd-Jones

et al 2019; Zhang et al 2018), LDpred-funct fits an infinitesimal model first and then reweighs the SNP effects by groups to approximate the sparse assumption. This two-stage weighting strategy is only an approximation of the mixture model approach, thus unlikely to surpass it.

3. For the trans-ancestry analyses, which LD matrix do SBayesR and SBayesRC use for the PRS calculation? Do you match the LD population to the GWAS samples (EUR) or the target samples?

We used the LD matrix from a random sample of 20K unrelated individuals of EUR ancestry in the UKB, when GWAS summary statistics from the EUR samples only were used for training the model for trans-ancestry prediction. In the analysis that combined data from the EUR sample and a non-EUR sample of the same ancestry with the target population, we ran SBayesRC in each sample separately using an ancestry-matched LD reference in the UKB. The principle is to use a LD reference that matches the GWAS sample in ancestry rather than the target sample. We have clarified this in the revised manuscript (lines 868-869 and 876-878).

4. In Figure 4, when incorporating more SNPs into the model in EAS samples, the performance of SBayesRC without annotation seems to be worse than SBayesR. Is there any possible explanation to this observation?

In the trans-ancestry prediction in EAS samples, the average prediction accuracy across traits was 0.058 from SBayesRC using 1M SNPs without annotations, which is equivalent to the standard SBayesR with the only difference in the way of regularization of LD matrix (a low-rank transformation is used in SBayesRC while a shrunk matrix is used in SBayesR). The average prediction accuracy from SBayesRC using 7M SNPs without annotations was also 0.058. While using 7M SNPs was not better in this case, the use of annotations or the combination of 7M SNPs and annotations increased the mean prediction accuracy to 0.062 and 0.065, respectively. This result suggests that annotations provide more information than high density SNPs for trans-ancestry prediction. This can possibly be explained by the combination of shared causal variants and LD differences between EUR and EAS samples. While including high-density SNPs would in principle help to select the causal variants in the model, the use of more SNPs alone may not be advantageous because there are more parameters to estimate from the data. A larger sample size is required to better estimate the variant effects, which is to some extent achieved using functional annotations. A related discussion on the added value of annotation data to the high-density SNP panel has been included in the revised manuscript (lines 508-516).

5. While trans-ancestry analyses aren't the focus of the current paper, it is still of interest to see the performance of SBayesRC vs PRS-CSx in African samples, which is known to have the worst portability in PRS analyses. Summary statistics from PAGE might be of use, though this might be something for a future paper.

We thank the reviewer for the suggestion and agree that it is of interest to compare SBayesRC vs PRS-CSx using PAGE summary statistics from mixed African samples. In the revised manuscript, we have included the analysis of 8 traits from the PAGE data set that were in common with those in the UKB data set of African ancestries. Similar to the analysis for EAS ancestry, we ran SBayesRC using UKB EUR data set only and SBayesRC-multi which combines data from both UKB EUR samples and PAGE samples, and compared to the performance of PRS-CSx. In SBayesRC-multi, we first ran SBayesRC in UKB EUR data set and PAGE data set separately and then used a tuning sample of 500 individuals of AFR ancestry in the UKB to generate African-specific SNP effect estimates. Likewise, we ran PRS-CSx using the same data with only UKB EUR data set for training or with UKB EUR and PAGE AFR data sets analysed jointly for training (with a tuning step). The prediction accuracy was calculated in the hold-out UKB AFR samples independent of the 500 tuning samples. As expected, in both SBayesRC and PRS-CSx, using both EUR and PAGE data sets for training was superior to using EUR data set only. However, SBayesRC with annotations using EUR data set only has readily outperformed PRS-CSx with both EUR and PAGE data as input, regardless of 1M or 7M SNPs used in SBayesRC (PRS-CSx cannot use 7M SNPs due to computational constraint). When SBayesRC-multi was used, with 7M SNPs, annotations, and AFR data from PAGE, the difference to PRS-CSx increased to 40.9% $\left(\frac{R_{S\text{BayesRC}}^2 - R_{\text{PRS-CSx}}^2}{R_{\text{PRS-CSx}}^2} \right)$ on average across traits, suggesting that information from annotations at dense SNPs can outweigh that from the summary statistics from the target population. These results are consistent with the observations from the trans-ancestry prediction for EAS (Figure 4c). We have included this new result in the revised Figure 4d, lines 354-359 in Results and lines 853-880 in Methods of the revised manuscript.

Some minor recommendation:

1. Considering the bottleneck of the current algorithm is due to the eigenvalue decomposition algorithm, a possible optimization for future version of SBayesRC might be the use of the Spectra C++ library, which allows for a quick calculation of eigen value and eigen vector on large sparse matrix.

We thank the reviewer for this suggestion. We found that Spectra C++ library is very efficient to generate a small number of top PCs but is not faster than Eigen C++ library when generating all eigenvalues and eigenvectors for a matrix, which is required in our method. We expect that general users would mostly likely only need to use the matrices provided on our website. If they need to perform eigen decomposition because the GWAS data is genetically different from our LD reference (unrelated UKB samples of European ancestry), the computation can be done in parallel between LD blocks and only needs to be run once for different traits. We will update the software when more efficient libraries become available in the future.

2. While I can find the R-version of the SBayesRC algorithm on the github page, it does not seem clear to me if the algorithm is implemented in the GCTB software. In the documentation on the GCTB website, while there is an option `--annot`, which I assume will run SBayesRC instead of SBayesR, the documentation stated that “The current version only allows binary annotations...” which seems to contradict to what was stated in the current paper, which stated that SBayesRC can handle continuous annotation. Maybe the authors should update the documentations of GCTB to reflect this change?

We have updated the GCTB software and the website to allow the use of SBayesRC with both quantitative and categorical annotations. The tutorial for SBayesRC can be found here: <https://cnsgenomics.com/software/gctb/#Tutorial>

3. A minor nitpicks on the coloring scheme: it would be nice if the coloring of each tools are consistent across different figures (e.g. LDpred2, SBayesR, SBayesRC, PolyPred-S).

We agree and have now used the same colour for each method across figures as possible as we can in the revised manuscript.

Reviewer #2:

Remarks to the Author:

The authors present an extension of the well established SBayesR methodology for PRS construction. This method adds functional labels to refine the SNP weights, which should in principle (and as shown here, in practice) improve fine-mapping, and therefore PRS robustness across ancestry groups.

As far as I can tell, SBayesR is used relatively widely and several other PRS prediction tools have developed their own version of functionally informed fine-mapping. It therefore makes sense to see SBayesR receive a similar upgrade. While the motivation and purpose and obviously well described in the literature, there are many useful insights from this paper, in particular the contribution of SNP density and its interaction with functional priors, together with added cross ancestry robustness. The analysis is thorough, exhaustive and rigorous. I cannot see major flaws in the arguments. The open source availability of the code is noted and much appreciated.

Reading the paper, my major interrogation lies in my limited understanding of the intuition behind the ability of the model to learn functional weights from the data. The principle of SBayesR, as far as I understand it, consists of reducing the complex SNP diversity using a low rank transformation that, in effect, collapses groups of variants in high LD into independent variables in the model. The idea behind

SNP functional data is of course the fact that even within an LD block, different variants can have very different functional annotations, and that should drive the choice of the variant receiving the higher weight. So I find it surprising that this low rank reduction can still enable the Bayesian machinery to learn about the different weights associated with functional annotations. Obviously it does, as evidenced in Figure 7c, but I would like the authors to help me understand why this training can work.

We thank the reviewer for the positive remarks and the comment on clarification. The reason why the Bayesian learning about functional annotations can still work in the low-rank model is, in brief, that the low rank approximation still captures >99% of the variance. However, it is expected that with a lower eigenvalue threshold (e.g., 50%), the performance would be compromised. A more detailed explanation is described as below.

The low-rank model is derived from the standard summary-data-based model, where m SNP marginal effects (\mathbf{b}) are fitted with m SNP joint effects ($\boldsymbol{\beta}$):

$$\mathbf{b} = \mathbf{R}\boldsymbol{\beta} + \boldsymbol{\varepsilon} \quad (1)$$

with \mathbf{R} being a $m \times m$ matrix and $\boldsymbol{\varepsilon}$ being mutually correlated with a variance-covariance matrix proportional to \mathbf{R} . In each LD block, we perform eigen-decomposition on \mathbf{R} such that

$$\mathbf{b} = \mathbf{U}\boldsymbol{\Lambda}\mathbf{U}'\boldsymbol{\beta} + \boldsymbol{\varepsilon} \quad (2)$$

where \mathbf{U} and $\boldsymbol{\Lambda}$ are matrices of eigenvectors and eigenvalues, respectively. The low-rank model is constructed such that the dimension of observed data (\mathbf{b} and \mathbf{R}) is reduced but the dimension of parameters ($\boldsymbol{\beta}$) remains. In essence, it fits q observables with m SNP joint effects, where $q \ll m$ and the observables are correlated only through SNP joint effects but not residuals:

$$\mathbf{w} = \mathbf{Q}\boldsymbol{\beta} + \boldsymbol{\epsilon} \quad (3)$$

with \mathbf{w} being a $q \times 1$ vector and \mathbf{Q} being a $q \times m$ matrix. As shown in our simulation and real data analyses, the low-rank model approximates the standard model with high accuracy, when the choice of q is well calibrated, and has considerable improvement in computational efficiency and robustness. The functional annotations are incorporated into the prior distribution of β_j :

$$\beta_j \sim \sum_k \pi_{jk} N(0, \sigma_k^2)$$

$$f(\pi_{jk}) = \mathbf{A}_j \boldsymbol{\alpha}_k$$

with \mathbf{A}_j being the observed annotation data matrix for SNP j and $\boldsymbol{\alpha}_k$ being the vector of unknown annotation effects estimated from the data. In this model, functional annotations describe the prior distribution of SNP joint effects, thereby contributing to the posterior estimation of SNP weights. It is a

hierarchical Bayesian learning process for α_k because α_k connect to the data (\mathbf{w} and \mathbf{Q}) only through β , which means that α_k can be estimated in either the standard model (1) or the low-rank model (3) since both models have β as parameters. Note that an alternative reduction is to reduce the dimension of parameter space, that is, from Eq (2) we can have

$$\mathbf{b} = \mathbf{U}\beta^* + \varepsilon$$

where \mathbf{U} is a $m \times q$ matrix and $\beta^* = \Lambda \mathbf{U}' \beta$ becomes a $q \times 1$ vector that captures the principal component effects. In this case, the Bayesian machinery to learn about the different weights associated with functional annotations would not work because each element of β^* is not the SNP joint effect itself but a linear combination of SNP joint effects. We have included this explanation in the Supplementary Note and mentioned at lines 156-158 in the revised manuscript.

A second comment – and I appreciate it may just be me – is that I am struggling to interpret performance measured as r^2 for binary traits. These are typically measured using AUC, Harrel's C or (better in my opinion) effect size per SD of the PRS (ideally conditional on age, sex and PCs). If I could encourage the authors to adapt ST3 to show these metrics for binary traits, that would be much appreciated.

We thank the referee for this question and for the suggestion. In the original manuscript, we used pseudo R^2 from the logistic regression that corrected for the covariates, including age, sex and PCs, to measure the prediction accuracy for binary traits. Unlike in case-control studies, the binary traits in this study were measured in the volunteered participants of UK Biobank, a sample from the general population.

Additionally, we used relative prediction accuracy for methods comparison $\left(\frac{R_{Full}^2 - R_{Basic}^2}{R_{Basic}^2} \right)$, where the effect of scale is cancelled so that the R^2 for quantitative traits and pseudo R^2 for binary traits are comparable. In the revised manuscript, we also calculated AUC and odds ratio per standard deviation of the PRS in the logistic regression conditional on the covariates, including age, sex and PCs, and reported these results in the revised Supplementary Table 3. Our understanding of Harrel's C-index is that it is commonly used to evaluate model performance in survival analysis but would be identical to AUC for binary data without time-to-disease measurements. In general, both AUC and odds ratio statistics for measuring prediction accuracy in diseases gave consistent results as shown by the pseudo R^2 . We have described in the main text at lines 829-833.

Related to this point of metrics for performance evaluation, the r^2 value associated with the PRS for height is massively affected by the inclusion of sex as a covariate in the model (because sex explains about 50% of the height variance, adding it as a covariate doubles the proportion of the residual variance explained by the PRS). I think this is what is stated at line 736 in the Methods section, but I would make it clear in the main text because, as far as height is concerned, this generates substantial confusion in the literature. I would double check that the statements on line 277 comparing the PRS variance to the theoretical variance explained by SNPs are not comparing oranges and apples (i.e. one includes sex as a

covariate and one does not). The large delta in these metrics between BMI and height makes me suspicious.

We confirmed that both prediction R^2 and SNP-based heritability estimates are based on data that have removed the mean and variance differences in sex by rank-based inverse-normal transformation of data within each sex group. The effects of age and 10 PCs were further removed from the GWAS summary data by fitting them as covariates in the GWAS model, and removed from the prediction evaluation by calculating an incremental R^2 of PRS through comparing the full model to the basic model that included these covariates. We have described these procedures in the main text at lines 743-745, 796-799, 825-826.

I would also tone down comments on the convergence issues observed in LDpred2 (line 180). These PRS tools typically require some minimum tuning and optimisation. Clearly, as authors of SBayes-RC, the authors will know how to make the most of the tool. My guess is that the LDpred2 authors (and I am not one of them!) would find a way to reach convergence. Hence, this comparison statement may come across as slightly unfair - though I appreciate that the authors also consider SBayesR, with extensive SBayesRC authorship overlap, so there is no doubt an argument for added robustness here.

We agree with the reviewer and have toned down the statement about LDpred2. Our revised manuscript now has the following statement (lines 187-188 “In an extreme case where African ancestry individuals were used to estimate LD correlations, ~70% prediction accuracy was preserved in SBayesRC, whereas both SBayesR and LDpred2 (under default settings) were unable to reach to convergence.”

I also would like to see the authors discuss the need for cross-validation in the UKB work. This is quite a substantial additional computational work, and the UKB is large enough to set aside a large training set while retaining excellent training sample size, especially for quantitative traits. Could I ask the authors to motivate that choice? Is that simply to extract as much data as possible from UKB or is there a rationale around added stability through that 10 fold procedure? Especially as the authors only use ~ 342K unrelated individuals, there are substantial numbers of EUR ancestry individuals that remain usable, and a minimum relatedness cutoff would yield a very large testing set with less than measurable bias associated with relatedness to training in my experience.

The k-fold cross-validation approach has been widely used in the PRS analysis. We reported the mean prediction R^2 across 10 folds and believe this is more reliable to the result from a single random sample for the training set with the remaining being the validation set, because the mean value is expected to average out potential ascertainment bias in partitioning training and validation samples. In addition, cross-validation provides a way to compute the standard error of the prediction accuracy for a trait, which is important when comparing differences in prediction performance between models/scenarios. For example, as shown in the revised Figure 5, the improvement in prediction accuracy due to the use of

annotations was significantly higher at 7M imputed SNPs than 1M HapMap3 SNPs for each trait, which can be seen from the standard error obtained from the cross-validation approach.

The reason we used 342K unrelated (genomic relationship cut-off at 0.05) individuals only is that SNPs can capture both direct and indirect genetic effects in the family as well as family-specific environmental effects, which will, if validation samples include relatives of training samples, lead to a higher prediction accuracy than that would be expected in a random sample of the population. Prediction accuracy in an independent random sample remains the most widely accepted criterion to assess the PRS performance. Therefore, we wished to only include the unrelated individuals in the validation set and did not use data from other individuals beyond the 342K unrelated samples of European ancestry.

We have included this discussion in the Supplementary Note.

I have a minor interpretation issue with the statement starting at line 361, in the results section. The authors make the case that annotation benefits will be maximised when working with sequence data. This seems contradictory with the fact that the authors see an asymptote in performance when a larger set of ~10M SNPs are used. The contradiction is resolved by arguing that this may be a limit of the imputation accuracy. That may be true but one could also argue in the other direction and say that the asymptote is reached because most causal variants are contained in the 7M set. In any case this is a discussion statement, not a result and I would be cautious with this as the added value of sequence data for PRS construction remains unclear.

We agree with the reviewer that this statement needs to be treated with caution. We have removed it from Results section and discussed this issue in the Discussion section (lines 518-530) with the following text.

“In the real data analysis, a significant improvement was observed from 1M to 7M imputed SNPs with annotations, but no further difference was observed with 10M imputed SNPs (Supplementary Fig. 20). This plateau in prediction performance could be attributed to the saturation of SNP tagging on the common causal variants by the 7M set or due to limitations in imputation accuracy on common SNPs or sample size of discovery GWAS, which is associated with the impact of MAF. This hypothesis can be tested when whole-genome sequence data for the full release of UKB samples becomes available. A recent study⁶² showed that 91% of rare variants identified using whole-exome sequence data in the UKB are independent of common variant signals. Furthermore, the imputation accuracy decreases with decreasing MAF⁶³, and the estimation variance of variant effects increases with MAF given a fixed sample size. Therefore, it is expected that using variants at sequence level, along with additional information from functional annotations, will provide added value for maximizing prediction accuracy.”

Just a minor technicality but my pdf version of Supp Tables only includes ST5. That confused me for a while until I found the remaining table in the excel doc. Something to check...

We thank the reviewer for a careful review. We have ensured all supplementary tables are shown correctly.

Vincent Plagnol

Reviewer #3:

Remarks to the Author:

This manuscript proposes a method SBayesRC to compute polygenic risk scores (PRS). SBayesRC extends SBayesR to use more SNPs and functional annotations. Simulation and real data analysis revealed that SBayesRC improves the prediction accuracy by leveraging the interaction between SNP density and functional annotation. Overall, I think SBayesRC is a solid method and the idea of using the low-rank model is clever. My main concern is about (1) the choice on how to model functional annotation with effect sizes and (2) lack of comparison with MegaPRS to show the improvement. Below are my comments.

We thank the reviewer for the positive remarks and constructive comments, which have helped us improve our method.

Major:

1. SBayesRC assumes that the probability of distribution membership π_{jk} is related to the functional annotation. Another way to model the functional annotation is to assume that the variance of effect sizes (heritability) for each SNP is additively related to the functional annotation (e.g. S-LDSC <https://www.ncbi.nlm.nih.gov/pmc/articles/PMC4626285/> and megaPRS <https://www.nature.com/articles/s41467-021-24485-y#Sec8>). The authors could discuss why they prefer to use the first approach to model the functional annotation.

We chose to model the functional annotation as a function of the probability of distribution membership because this model is straightforward to interpret, especially in the context of a multi-component mixture model, as explained below.

S-LDSC assumes that each SNP effect has a univariate normal distribution and is not from a mixture model. In this case, it makes perfect sense to model the SNP effect variance as a weighted sum of components with respect to functional annotations, i.e., $\beta_j \sim N(0, \sigma_j^2)$, with $\sigma_j^2 = \sum_c a_{jc} \tau_c^2$ where a_{jc} is the value of annotation c at SNP j and τ_c^2 is the heritability enrichment in annotation c . MegaPRS is a tool comprising multiple summary-data-based prediction methods, including an infinitesimal model (LDAK-Ridge-SS), a two-normal mixture model (LDAK-Bolt-SS), and a multi-component mixture model (LDAK-BayesR-SS). LDAK-Ridge-SS can be regarded as an equivalent model as S-LDSC. However, in LDAK-BayesR-SS, $\beta_j \sim \pi_1 \delta_0 + \pi_2 N(0, \sigma_j^2/100) + \pi_3 N(0, \sigma_j^2/10) + \pi_4 N(0, \sigma_j^2)$, the biological interpretation is not straightforward. For example, π_2 cannot be interpreted as the proportion of SNPs in the small effect size category because whether the effect size is small or large depends on σ_j^2 which is annotation dependent. This may potentially affect the identifiability of the mixture membership if annotation-specific heritability enrichment parameters τ_c^2 are estimated jointly with the SNP effects β_j (instead, a two-step estimation approach is used in the MegaPRS paper, where τ_c^2 are estimated by BLD-LDAK model first and then treated as unknown quantities in LDAK-BayesR-SS).

To improve the model interpretability and potentially identifiability and robustness, we assume that

$$\beta_j \sim \pi_{j1} \delta_0 + \pi_{j2} N(0, \sigma_g^2/10000) + \pi_{j3} N(0, \sigma_g^2/1000) + \pi_{j4} N(0, \sigma_g^2/100)$$

where the mixture distributions are independent of annotations, which represent zero, small, medium and large effect sizes that, in order, explains 0, 0.01%, 0.1% and 1% of the total genetic variance (σ_g^2). The mixture distribution membership probabilities, $\pi_{j1}, \pi_{j2}, \pi_{j3}, \pi_{j4}$, are dependent of annotations such that the interpretation for the annotation effects are straightforward, that is, the higher value of the annotation would increase or decrease the probability of the SNP with an effect belong to a given effect size distribution. Since the model is clearly defined, there is no identifiability problem when estimating SNP effects and annotation effects altogether, which we believe is favourable compared to the two-stage estimation approach.

From the simulation and real data analysis, SBayesRC gives higher prediction accuracy than MegaPRS (Fig. 3b and Supplementary Fig. 14 in the revised manuscript), supporting that the first approach is likely a better way of modelling functional annotations than the second approach. We have included this discussion in the Supplementary Note.

2. At the same point, for the simulation scenario incorporating annotation data, the SNP probability of membership was linked with the annotation. It would be helpful to investigate the performance of SBayesRC when the variance of effect sizes is additively linked to the functional annotation.

We have performed additional simulations where the variance of effect sizes is additively linked to the functional annotations (similar to the LDAK-BayesR-SS or S-LDSC model). The data generative model is:

$$\beta_j \sim \pi_1 \delta_0 + \pi_2 N(0, \sigma_j^2/100) + \pi_3 N(0, \sigma_j^2/10) + \pi_4 N(0, \sigma_j^2)$$

where $\sigma_j^2 = \sum_c a_{jc} \tau_c^2$ with a_{jc} being the observed annotation per SNP and τ_c^2 being the annotation heritability enrichment parameter for height, and $\pi_1, \pi_2, \pi_3, \pi_4$ are set to be 0.998, 1.89e-3, 9.26e-6, and 1.97e-9, respectively. The trait heritability was set to be 0.1 or 0.5. The pattern of prediction accuracy from using 1M or 7M SNPs and with or without annotations in SBayesRC is similar between the two data generative models (SBayesRC model or S-LDSC model for simulation). In both cases, SBayesRC with 7M SNPs and annotations gave the highest prediction accuracy. We noted that overall the prediction accuracy using the S-LDSC model in the simulation was higher than that using the SBayesRC model in the simulation. This may be because, under the S-LDSC model, some causal variants can have a very large causal effect, leading to a large discovery power from GWAS, and therefore higher prediction accuracy. In this case, the added value from the annotations became trivial.

We compared the performance of SBayesRC and MegaPRS (LDAK-BayesR-SS) model and found that regardless of which data generative model was used, SBayesRC always gave a higher prediction accuracy than MegaPRS. The superiority of SBayesRC may result from three reasons. First, modelling annotations as a function of pi values may be a better way of incorporating annotation data, as explained above. Second, in SBayesRC, all parameters are estimated coherently in one model, whereas MegaPRS takes heritability enrichment estimates and different sets of pi values as input and do not consider the estimation variance for these parameters. Third, SBayesRC simultaneously fits all SNPs in the model, but MegaPRS only fits a subset of SNPs that are enriched in per-SNP heritability comparing to a random SNP in the genome. SNPs with depletion in heritability enrichment may still have a nonzero, albeit small, effect on the trait, and completely ignoring them may adversely affect prediction accuracy.

We have included this result in Supplementary Figure 14 and in the main text at lines 230-234.

3. SBayesRC assumes that the scaling factor of five distributions as the mixture components is fixed as $\gamma = [0, 0.001, 0.01, 0.1, 1]$. In simulation, the authors specified the correct γ . I was wondering whether SBayesRC would be robust to the misspecification of γ .

For the simulated data, we compared the SBayesRC analysis with the correct gamma values used in the simulation, i.e., $\gamma = [0, 0.001, 0.01, 0.1, 1]$, to the analysis with a very different set of gamma values, $\gamma = [0, 0.001, 0.005, 0.01, 0.02]$. The result showed that the difference in prediction accuracy is negligible, suggesting that SBayesRC is robust to the misspecification of the gamma values

(Supplementary Figure 12). We have included this in main text lines 227-230, and supplementary Figure 12.

4. To my knowledge, SBayesR is not the first method that utilizes the functional annotation and use more SNPs to compute PRS. MegaPRS also uses functional annotation, ~7.5M imputed SNPs (<https://www.nature.com/articles/s41467-021-24485-y>) and has similar assumptions on effect sizes (multi-component mixture priors). It would be helpful if the authors could compare SBayesRC with MegaPRS in simulations and real data analysis.

We thank the reviewer for raising the point that MegaPRS is also capable of analysing 7M SNPs with annotations so we should compare the two methods in simulations and real data analyses. To investigate this, we have done a set of additional analyses, using MegaPRS, in particular, the LDAK-BayesR-SS model (multi-component mixture prior), with 7M SNPs and functional annotations. First, we simulated data based on the SBayesRC model and analysed using the SBayesRC or MegaPRS methods. Second, we simulated data based on the MegaPRS model (or equivalently, the LDSC model, when no extra LD weights are modelled) and analysed using the SBayesRC or MegaPRS methods. The details of simulation are described in our response to the reviewer's comment #2. We found that SBayesRC gave a higher prediction accuracy than MegaPRS regardless of which data generative model was used in the simulation (new Supplementary Figure 14). Third, we ran MegaPRS in the 10-fold cross-validation analysis in the UKB of European ancestry, and found that, although outperformed LDpred2 and LDpred-funt, the prediction accuracy was lower than that from SBayesRC (revised Fig. 3a), consistent with the simulation results. Furthermore, we applied MegaPRS in cross-biobank prediction analyses, including training in FinnGen and validation in UKB, and training in the data set from published meta-analysis and validation in Lifelines, across quantitative traits and diseases. Our results showed that SBayesRC had a better predictive performance than MegaPRS, measured by the prediction accuracy relative to LDpred2 (we did not compute the prediction accuracy relative to the standard SBayesR because of the convergence issue in the standard SBayesR for some of the traits) (revised Fig. 3b). Additionally, the advantage of SBayesRC over MegaPRS is observed with different training sample sizes for height and BMI (revised Fig. 3c). Moreover, our investigation in this study identified a significant interaction between SNP density and annotation information for improving polygenic prediction. We observed this beneficial interactive effect in both within-EUR and cross-ancestry prediction. We note that this is an important result that has not been concluded by the MegaPRS paper, which mainly focused on ~600K directly genotyped SNPs. Last but not least, although MegaPRS is capable of analysing 7M SNPs with annotations, it requires almost 3x larger memory than our method (revised Table 2).

We have included these new results in Fig. 2c, 3, Supplementary Fig. 14, the Supplementary Note, and the Results section in the revised manuscript at lines 214-217 and 267-271, Methods section at lines 812-820.

5. In my opinion, out-of-sample prediction performance based on external summary statistics is the most important and commonly used evaluation approach for the benchmark of PRS methods. Based on my experience, SBayesR can sometimes have convergence problems when applied to external summary statistics. Currently only 2 traits (height and BMI) were analyzed in the benchmark. It would be more convincing if the authors could demonstrate that SBayesRC outperforms other methods on more traits using external summary statistics.

We thank the reviewer for the comment, which has helped us to identify a potential problem of our method and has led to a methodological improvement.

We aimed to run SBayesRC in more traits using external summary statistics. However, the most recently published GWAS for complex traits are predominantly based on the UKB data or have included UKB as part of the meta-analysis. Therefore, the number of applicable traits is limited to the data availability in Lifelines, our independent validation cohort. We found 2 more traits that have a large GWAS sample size and a decent validation sample size in Lifelines, resulting in a total of 4 traits: height (GWAS data from Yengo et al; $n_{\text{GWAS}}=704,823$; $n_{\text{Lifelines}}=11,842$), BMI (Yengo et al; $n_{\text{GWAS}}=688,633$; $n_{\text{Lifelines}}=11,842$), diastolic blood pressure (Evangelou et al; $n_{\text{GWAS}}=756,595$; $n_{\text{Lifelines}}=11,832$), and T2D (Xue et al; $n_{\text{cases_GWAS}}=62,693$; $n_{\text{cases_Lifelines}}=179$). To evaluate our method in more traits, we performed another set of analysis using the summary statistics from the FinnGen data set (max $n=342,499$) for training, with the 350K unrelated UKB individuals of European ancestry as validation. We selected 5 common diseases in FinnGen that have at least 1,000 cases in the UKB. Additionally, we have analysed the summary statistics from 8 traits in Biobank Japan using the UKB EAS samples as LD reference, and another 8 traits in PAGE using the UKB AFR samples as LD reference. Put all together, we have analysed 25 sets of external summary statistics from traits across a range of categories (Supplementary Tables 5-7). The results of these analyses are shown in the revised Figures 3b and Figure 4c,d. Our results showed that SBayesRC outperformed LDpred2, LDpred-funct and MegaPRS in out-of-sample prediction and outperformed PRS-CSx in cross-ancestry prediction, when external summary statistics were used for training.

In our analysis of FinnGen summary statistics, we used the UKB EUR samples as LD reference because the individual genotype data for the FinnGen population are not publicly available. In this case, we observed a convergence problem in some of the traits analysed, when using our default settings in SBayesRC. Through investigation, we found the problem to be related to the specification of the cut-off value (ρ) for the variance explained in LD by the selected eigenvalues. The default value of ρ was set to be 0.995, based on the simulation study using the UKB data. However, for some traits in FinnGen, a lower ρ value was required to address the convergence in SBayesRC. The intuitive explanation is that when the GWAS sample is different from the reference sample in LD structure (e.g., FinnGen is an

isolated population), the LD matrix computed from the reference requires stronger regularization, which can be achieved by only including principal components with relatively large eigenvalues, since those with small eigenvalues are subject to LD variation. In other words, the ρ value that optimises the prediction performance is data set dependent, particularly contingent on the LD differences between GWAS and reference samples. We have now improved our method to automatically choose the optimal ρ value by performing a pseudo validation based on the summary statistics, following the strategy of Zhang et al (the MegaPRS paper). In brief, we added a tuning step which will partition the observed summary statistics into training and test sets, run multiple short MCMC chains with different ρ values, and then use the one that gives the highest prediction accuracy in the test set for the formal analysis. In contrast to Zhang et al, our approach for partitioning the summary statistics does not require individual genotype data, but only eigenvalues and eigenvectors of the LD matrix, which have already been generated for the low-rank model. Additionally, it is of note that this tuning step is only used to find the optimal ρ value and all the other model parameters remain estimated from the data. The details of this new component of our method are described in the Supplementary Note and the main text at lines 173-176 and 643-649.

6. Have the authors considered using AFR instead of EAS summary statistics (e.g. PAGE) to derive PRS for trans-ancestry prediction in AFR populations?

In the revised manuscript, we have included the analysis of 8 traits from the PAGE data set that were in common with those in the UKB data set of African ancestries. Similar to the analysis for EAS ancestry, we ran SBayesRC using UKB EUR data set only and SBayesRC-multi which combines data from both UKB EUR samples and PAGE samples, and compared to the performance of PRS-CSx. In SBayesRC-multi, we first ran SBayesRC in UKB EUR data set and PAGE data set separately and then used a tuning sample of 500 individuals of AFR ancestry in the UKB to generate African-specific SNP effect estimates. Likewise, we ran PRS-CSx using the same data with only UKB EUR data set for training or with UKB EUR and PAGE AFR data sets analysed jointly for training (with a tuning step). The prediction accuracy was calculated in the hold-out UKB AFR samples independent of the 500 tuning samples. As expected, in both SBayesRC and PRS-CSx, using both EUR and AFR data sets for training was superior to using EUR data set only. However, SBayesRC with annotations using EUR data set only has readily outperformed PRS-CSx with both EUR and AFR data as input, regardless of 1M or 7M SNPs used in SBayesRC (PRS-CSx cannot use 7M SNPs due to computational constraint). When SBayesRC-multi was used, with 7M SNPs, annotations, and AFR data from PAGE, the difference to PRS-CSx increased to 40.9%

$\left(\frac{R_{\text{SBayesRC}}^2 - R_{\text{PRS-CSx}}^2}{R_{\text{PRS-CSx}}^2} \right)$ on average across traits, suggesting that information from annotations at dense SNPs can outweigh that from the summary statistics from the target population. These results are consistent with the observations from the trans-ancestry prediction for EAS (Figure 4c). We have included this new result in the revised Figure 4d, lines 354-359 in Results and lines 853-880 in Methods of the revised manuscript.

Minor:

1. In Figure 1a, why is Anno3 of "causal variant unobserved" different with Anno3 of "causal variant observed"?

We have improved Figure 1a to better convey the idea that when the causal variant is not observed, its effect can be captured through LD by a SNP that has a different annotation from the causal variant.

2. Results from PRS-CSx should be included in Figure 4.

The results from PRS-CSx have already been shown in Figure 4c and 4d (the first column). We have made this clear in the figure legend.

3. SBayesRC uses 5 mixture components. I was wondering whether increasing or decreasing the number of components could affect the performance.

The impact of the number of mixture component on the prediction accuracy has been explored in the SBayesR paper (Lloyd-Jones et al). In the revised manuscript, we run SBayesRC with 4 or 6 mixture components (default is 5) in the simulation, which was based on 5 components, as well as in one quantitative trait (height) and one disease trait (T2D) and found negligible difference to the default setting of 5 components. This result is now shown in Supplementary Figure 13 and in the main text at lines 227-230.

4. Tuning parameters of LDpred2 and PRS-CSx should be listed in the method section.

We have added these details in the revised manuscript (lines 814-817, 824-827 and 870-875).

5. In Figure 7c, what does the x-axis of "zero effect" mean? Why do some categories have negative values?

Figure 7c shows the deviation of proportion of SNPs in each effect size distribution (from zero, small to very large effect sizes) for a functional category to the overall proportion of genome-wide SNPs across functional categories. We have made this clear in the figure legend (lines 1268-1270)

Decision Letter, first revision:

23rd June 2023

Dear Jian,

Your revised Analysis "Leveraging functional genomic annotations and genome coverage to improve polygenic prediction of complex traits within and between ancestries" has been seen by two of the original referees. You will see from their comments below that, while they find the manuscript improved, they have highlighted a few ongoing concerns. We remain interested in the possibility of publishing your study in Nature Genetics, but we would like to consider your response to these concerns in the form of a further revision before we make a final decision on publication.

As before, to guide the scope of the revisions, the editors discuss the referee reports in detail within the team, including with the chief editor, with a view to identifying key priorities that should be addressed in revision, and sometimes overruling referee requests that are deemed beyond the scope of the current study. In this case, we ask that you revise the figures for clarity as requested by Reviewer #1 and extend the benchmarking analyses where feasible as requested by Reviewer #3. We again hope that you will find this prioritized set of referee points to be useful when revising your study. Please do not hesitate to get in touch if you would like to discuss these issues further.

We therefore invite you to revise your manuscript again taking into account all reviewer and editor comments. Please highlight all changes in the manuscript text file. At this stage, we will need you to upload a copy of the manuscript in MS Word .docx or similar editable format.

*2) If you have not done so already, please begin to revise your manuscript so that it conforms to our Analysis format instructions, available here. Refer also to any guidelines provided in this letter.

Please be aware of our guidelines on digital image standards.

[redacted]

We hope to receive your revised manuscript within 4-8 weeks. If you cannot send it within this time, please let us know.

Sincerely,
Kyle

Kyle Vogan, PhD
Senior Editor
Nature Genetics
<https://orcid.org/0000-0001-9565-9665>

Referee expertise:

Referee #1: Genetics, statistical methods, polygenic prediction

Referee #3: Genetics, statistical methods, polygenic prediction

Reviewers' Comments:

Reviewer #1:
Remarks to the Author:

In this update, the authors have included additional analyses, incorporating MegaPRS and external GWAS for different ancestries such as African (PAGE). This has greatly improved the current paper and has consistently demonstrated the performance gain of SBayesRC. While the content of the current paper has greatly improved, the figure illustrations can be improved. Specifically, I have the following comments:

1. The only question I have, which is more for my own curiosity, with regarding to the performance of SBayesRC is whether the inclusion of "junk" annotations will have a negative impact on SBayesRC? For example, if a random set of annotations were added, will performance of SBayesRC reduce? This is not major, and is really only for my own curiosity.
2. I struggle to find the exact analyses that was done to generate Figure 2. Based on the description in the method section, it was suggested that traits with heritability of 0.1 and 0.5 were both simulated. However, looking at Figure 2a and Figure 2b, it's likely that only the results from heritability of 0.5 were reported? Would it be possible to highlight that in the legend, or with a horizontal dotted line (with description / legend) to show what's the "expected" prediction accuracy under the simulation?
3. For Figure 2c, the legend stated that it was the improvement of prediction accuracy using SBayesRC comparing to other methods, but there are boxes with color of SBayesRC. This raises two questions: does the y-axis show the relative difference in prediction accuracy (e.g. R^2 from (SBayesRC - R^2 from X) / R^2 from X, where X are the different methods), or does it show the absolute difference in R^2 ? But for either, it is rather strange that SBayesRC is included in the figure. Or did the authors intend to show the absolute / relative difference between the methods and SBayesR instead? And what does the dotted horizontal line represent?
4. For Figure 2d-f, it is difficult to differentiate the alpha change between the annotation classes, maybe it's better if a different shade of color is used? e.g. dark red vs pink (or #D81B60 vs #FFC107 for colorblind friendly contrast).
5. Similarly, the color palette for Figure 3 is rather difficult to read. Maybe you can try to use some of the palette provided by ggsci package instead of the ggplot default to make it slightly easier.

Minor comment:

1. Figure 4's legend, B should be b instead

Reviewer #3:

Remarks to the Author:

I thank the authors for addressing the majority of my previous comments. I have also tested SBayesRC myself and found that SBayesRC converges on external summary statistics that SBayesR is unable to. I have two more questions.

My main concern is that the improvement of SBayesRC over MegaPRS is marginal (4.1 %). Perhaps

the authors could analyze more traits to more comprehensively demonstrate the improvement of SBayesRC. For example, MegaPRS can be run in the same way as SBayesRC for the comparison of trans-ancestry prediction (Figure 4).

In the simulation setting for comparison of MegaPRS, the effect size β_j was simulated from $\pi_1 \delta_0 + \pi_2 N(0, \sigma_j^2/100) + \pi_3 N(0, \sigma_j^2/10) + \pi_4 N(0, \sigma_j^2)$. I think the correct way is to simulate β_j from $\pi_1 \delta_0 + \pi_2 N(0, s \sigma_j^2/100) + \pi_3 N(0, s \sigma_j^2/10) + \pi_4 N(0, s \sigma_j^2)$ where $s = (\pi_2/100 + \pi_3/10 + \pi_4)^{-1}$ as the current form does not guarantee the heritability of SNP j is σ_j^2 .

Additional Review Material, First Revision

Dear Jian,

We just received an additional review from Reviewer #2 (copied below). When preparing your revision, we ask that you address these comments and include this third review in your point-by-point response. Thanks in advance!

Best,
Kyle

Reviewer #2 (Remarks to the Author):

First of all, I apologise to the authors for this late review - I somehow did not receive reminders and this review was lost in my inbox for some time.

Now about the revised paper, I appreciated the efforts put to address the comments from all the reviewers. I read with interest the section on the low rank model and its impact on annotation effects. While quite technical, I did learn something useful.

I do think this paper is ready for publication. A last question I have relates to the availability of the optimised PGS, typically in the PGS catalog. It would be helpful to do so for the traits presented in Figure 3 but the data availability section makes no mention of this. Do the authors plan to change this?

Author Rebuttal, first revision:

We thank the reviewers for their positive remarks and additional constructive suggestions. We have improved the clarity of figures, made our PGS predictors publicly available, and extended the benchmarking analysis accordingly. Please find our point-to-point response to the reviewer's comments in blue.

Reviewers' Comments:

Reviewer #1:

Remarks to the Author:

In this update, the authors have included additional analyses, incorporating MegaPRS and external GWAS for different ancestries such as African (PAGE). This has greatly improved the current paper and has consistently demonstrated the performance gain of SBayesRC. While the content of the current paper has greatly improved, the figure illustrations can be improved. Specifically, I have the following comments:

1. The only question I have, which is more for my own curiosity, with regarding to the performance of SBayesRC is whether the inclusion of “junk” annotations will have a negative impact on SBayesRC? For example, if a random set of annotations were added, will performance of SBayesRC reduce? This is not major, and is really only for my own curiosity.

We shared the same curiosity with the reviewer and have investigated the performance of SBayesRC by simulating random numbers (random samples from a uniform distribution between zero and one) for each of the annotations used in the simulation. The prediction accuracy using random annotations has been shown in the Supplementary Figure 11 in the previous version. The result showed that the inclusion of random annotations in SBayesRC did not adversely affect prediction when the trait heritability was 0.5 but slightly decreased the prediction accuracy (by 2%) when the trait heritability was 0.1. Our further investigation revealed that the slight decrease in prediction accuracy in the case of low heritability is likely due to the increased dispersion of annotation effect estimates around zero when the power of estimation is relatively low (shown in the new panel b of the Supplementary Figure 12 in the revised manuscript). Therefore, we recommend use informative annotations especially for traits with limited GWAS power (e.g., low SNP-based heritability or limited GWAS sample size). We have included the new result (**Supplementary Figure 12b**) and discussed in lines 594-597.

2. I struggle to find the exact analyses that was done to generate Figure 2. Based on the description in the method section, it was suggested that traits with heritability of 0.1 and 0.5 were both simulated. However, looking at Figure 2a and Figure 2b, it's likely that only the results from heritability of 0.5 were reported? Would it be possible to highlight that in the legend, or with a horizontal dotted line (with description / legend) to show what's the “expected” prediction accuracy under the simulation?

We have improved the figure legend as following. “Results were from simulations with trait heritability $h^2=0.5$ (the upper bound of the prediction accuracy). See **Supplementary Fig. 4-8** for results from the simulation with $h^2=0.1$.”

3. For Figure 2c, the legend stated that it was the improvement of prediction accuracy using SBayesRC comparing to other methods, but there are boxes with color of SBayesRC. This raises two questions: does the y-axis show the relative difference in prediction accuracy (e.g. R^2 from (SBayesRC – R^2 from X) / R^2 from X, where X are the different methods), or does it show the absolute difference in R^2 ? But for either, it is rather strange that SBayesRC is included in the figure. Or did the authors intend to show the absolute / relative difference between the methods and SBayesR instead? And what does the dotted horizontal line represent?

We apologise for the confusion caused by the misleading figure legend. We have revised the figure legend as below. c) The prediction R-squared from SBayesRC, LDpred-funct, and MegaPRS with different SNP densities and with or without annotations. The dashed line shows the prediction R-squared from the benchmarking method SBayesR using HapMap3 SNPs without annotations. Results were from simulations with trait heritability $h^2=0.5$ (the upper bound of the prediction accuracy). See **Supplementary Fig. 4-8** for results from the simulation with $h^2=0.1$.

4. For Figure 2d-f, it is difficult to differentiate the alpha change between the annotation classes, maybe it's better if a different shade of color is used? e.g. dark red vs pink (or #D81B60 vs #FFC107 for colorblind friendly contrast).

We have updated Figure 2d-f accordingly.

5. Similarly, the color palette for Figure 3 is rather difficult to read. Maybe you can try to use some of the palette provided by ggsci package instead of the ggplot default to make it slightly easier.

We have tried other packages but didn't find a palette that has a significantly better colour contrast. To be consistent across figures, we therefore prefer to keep the current palette in the final version.

Minor comment:

1. Figure 4's legend, B should be b instead

We have fixed this typo.

Reviewer #2 (Remarks to the Author):

First of all, I apologise to the authors for this late review - I somehow did not receive reminders and this review was lost in my inbox for some time.

Now about the revised paper, I appreciated the efforts put to address the comments from all the reviewers. I read with interest the section on the low rank model and its impact on annotation effects. While quite technical, I did learn something useful.

I do think this paper is ready for publication. A last question I have relates to the availability of the optimised PGS, typically in the PGS catalog. It would be helpful to do so for the traits presented in Figure 3 but the data availability section makes no mention of this. Do the authors plan to change this?

We thank the referee for the positive comments. The summary data from GWAS and SNP weights from SBayesRC for constructing the PGS in **Fig. 3** can now be found on the GCTB website. We have reservations about directly depositing the SNP weights from SBayesRC to the PGS catalog as the PGS catalog is limited to the additional information that can be provided to the users. Compared to most repositories in the PGS catalog that consist of only a subset of SNPs with significant marginal effects, our SNP weights are joint effects derived from genome-wide SNPs. In this case, it's important to have matched SNP set between training and validation datasets. This is because if some important SNPs present in the training are missing in the validation, the genetic effects captured by these SNPs will be lost. To maximise the utility of the joint SNP weights, we have included a text on our website for cautions and recommendations of use, as well as the GWAS summary data used in this study for these traits, considering that the users may need to rerun SBayesRC with matched SNPs if significant proportions of SNPs were missing from their target population of prediction. We have included in the data availability lines 921-924.

Reviewer #3:

Remarks to the Author:

I thank the authors for addressing the majority of my previous comments. I have also tested SBayesRC myself and found that SBayesRC converges on external summary statistics that SBayesR is unable to. I have two more questions.

My main concern is that the improvement of SBayesRC over MegaPRS is marginal (4.1 %). Perhaps the authors could analyze more traits to more comprehensively demonstrate the improvement of SBayesRC. For example, MegaPRS can be run in the same way as SBayesRC for the comparison of trans-ancestry prediction (Figure 4).

We thank the referee for the positive comments and the suggestions. Indeed, SBayesRC has only a small improvement compared to MegaPRS when the LD reference is the same as or closely related to the

GWAS sample. However, as shown in Figure 3b of the previous manuscript, SBayesRC gave significantly higher prediction accuracy than MegaPRS across different traits and diseases when using an external LD reference that was independent of the GWAS sample. In the revised manuscript, we have further included the results of analyses within populations of non-European ancestries using external LD reference (specifically, analysis of BBJ data using UKB EAS as LD reference, and analysis of PAGE data using UKB AFR as LD reference) and run MegaPRS in the same way as SBayesRC for trans-ancestry prediction. We found that the benefit of SBayesRC over MegaPRS in this case held in different ancestries as well as in trans-ancestry prediction when EUR and target ancestry data sets were combined to improve prediction. The results are as described in detail below.

1) For within-ancestry prediction, SBayesRC had a significant advantage over MegaPRS when using an external LD reference independent of the GWAS sample. We have previously shown that SBayesRC led to 29.2% higher prediction accuracy than MegaPRS, on average across 5 common diseases, when GWAS summary data from FinnGen was used for training, a random sample of UKB EUR as LD reference, and the full UKB EUR as validation. The significant improvement is likely because through the low-rank approximation SBayesRC better accounted for the sampling variation in LD between the reference and GWAS samples, as a substantial sampling variation is likely to exist between FinnGen and UKB populations, considering Finnish to be an isolated EUR population according to the demographic history. We further evaluated the performance of SBayesRC and MegaPRS in the case of LD sampling variation in EAS and AFR ancestries. For EAS ancestry, we used GWAS summary data from BBJ as training, UKB EAS as LD reference, and UKB EAS as validation. Similarly, for AFR ancestry, we used GWAS summary data from PAGE as training, UKB AFR as LD reference, and UKB AFR as validation. Compared to MegaPRS, SBayesRC led to an improvement of 4.0% and 39.0% prediction accuracy, across 8 traits in EAS and AFR ancestries respectively (**Supplementary Fig. 19**). The improvement in AFR ancestry was the largest, because there are larger LD differences between UKB AFR and PAGE which has a mixed African ancestry.

2) For trans-ancestry prediction, SBayesRC-multi that combines UKB EUR data set with PAGE data set gave significantly better prediction accuracy than MegaPRS-multi, on average across traits (mean relative prediction accuracy of 38.7% from SBayesRC-multi vs. that of 26.6% from MegaPRS-multi). In addition, the variation in prediction accuracy across traits was smaller in SBayesRC-multi compared to MegaPRS-multi. The benefit of SBayesRC-multi is mainly because the PAGE data was better used in SBayesRC than in MegaPRS, as described above.

Lastly, as per the reviewer's suggestion, we also ran trans-ancestry prediction in UKB using EUR as training and other ancestries as validation. The results are in general consistent with the within ancestry prediction that SBayesRC has a slightly higher mean prediction accuracy and smaller variance than MegaPRS. In these cases, LD reference was from the same population as the GWAS samples. As described above, the benefit of SBayesRC mainly lies at the circumstances where only an external LD reference is available for the analysis.

We have included the new results in **Fig. 4**, **Supplementary Fig. 19**, results at lines 275-281, 338-353, and discussed at lines 542-547 in the revised manuscript.

In the simulation setting for comparison of MegaPRS, the effect size β_j was simulated from $\pi_1 \delta_0 + \pi_2 N(0, \sigma_j^2/100) + \pi_3 N(0, \sigma_j^2/10) + \pi_4 N(0, \sigma_j^2)$. I think the correct way is to simulate β_j from $\pi_1 \delta_0 + \pi_2 N(0, s \sigma_j^2/100) + \pi_3 N(0, s \sigma_j^2/10) + \pi_4 N(0, s \sigma_j^2)$ where $s = (\pi_2/100 + \pi_3/10 + \pi_4)^{-1}$ as the current form does not guarantee the heritability of SNP j is σ_j^2 .

We thank the referee for capturing this. We have simulated using the exactly the same model as described above and rerun the analysis. As shown in the new **Supplementary Fig. 15**, the results did not change much and the conclusion remains the same. We have revised the text of the Supplementary Note in page 17 accordingly.

Decision Letter, second revision:

30th October 2023

Dear Jian,

Thank you for submitting your revised manuscript "Leveraging functional genomic annotations and genome coverage to improve polygenic prediction of complex traits within and between ancestries" (NG-AN61070R1). In light of the changes made in response to the referees' comments, we will be happy in principle to publish your study in Nature Genetics as an Analysis pending final revisions to comply with our editorial and formatting guidelines.

We are now performing detailed checks on your paper, and we will send you a checklist detailing our editorial and formatting requirements soon. Please do not upload the final materials or make any revisions until you receive this additional information from us.

Thank you again for your interest in Nature Genetics. Please do not hesitate to contact me if you have any questions.

Sincerely,
Kyle

Kyle Vogan, PhD
Senior Editor
Nature Genetics
<https://orcid.org/0000-0001-9565-9665>

Final Decision Letter:

In reply please quote: NG-A61070R2 Zeng

5th March 2024

Dear Jian,

I am delighted to say that your manuscript "Leveraging functional genomic annotations and genome coverage to improve polygenic prediction of complex traits within and between ancestries" has been accepted for publication in an upcoming issue of Nature Genetics.

Your paper will be published online after we receive your corrections and will appear in print in the next available issue. You can find out your date of online publication by contacting the Nature Press Office (press@nature.com) after sending your e-proof corrections.

Before your paper is published online, we will be distributing a press release to news organizations worldwide, which may very well include details of your work. We are happy for your institution or funding agency to prepare its own press release, but it must mention the embargo date and Nature Genetics. Our Press Office may contact you closer to the time of publication, but if you or your Press Office have any enquiries in the meantime, please contact press@nature.com.

Please note that Nature Genetics is a Transformative Journal (TJ). Authors may publish their research with us through the traditional subscription access route or make their paper immediately open access through payment of an article-processing charge (APC). Authors will not be required to make a final decision about access to their article until it has been accepted. Find out more about Transformative Journals

Authors may need to take specific actions to achieve compliance with funder and institutional open access mandates. If your research is supported by a funder that requires immediate open access (e.g. according to Plan S principles), then you should select the gold OA route, and we will direct you to the compliant route where possible. For authors selecting the subscription publication route, the journal's standard licensing terms will need to be accepted, including [a href="https://www.nature.com/nature-portfolio/editorial-policies/self-archiving-and-license-to-publish](https://www.nature.com/nature-portfolio/editorial-policies/self-archiving-and-license-to-publish). Those licensing terms will supersede any other terms that the author or any third party may assert apply to any version of the manuscript.

If you have not already done so, we invite you to upload the step-by-step protocols used in this manuscript to the Protocols Exchange, part of our on-line web resource, natureprotocols.com. If you complete the upload by the time you receive your manuscript proofs, we can insert links in your article that lead directly to the protocol details. Your protocol will be made freely available upon publication of your paper. By participating in natureprotocols.com, you are enabling researchers to more readily reproduce or adapt the methodology you use. [Natureprotocols.com](http://natureprotocols.com) is fully searchable, providing your

protocols and paper with increased utility and visibility. Please submit your protocol to <https://protocolexchange.researchsquare.com/>. After entering your nature.com username and password you will need to enter your manuscript number (NG-A61070R2). Further information can be found at <https://www.nature.com/nature-portfolio/editorial-policies/reporting-standards#protocols>

Sincerely,
Kyle

Kyle Vogan, PhD
Senior Editor
Nature Genetics
<https://orcid.org/0000-0001-9565-9665>